# A Comparison of Depression and Anxiety among University Students in Nine Countries during the COVID-19 Pandemic

**DOI:** 10.3390/jcm10132882

**Published:** 2021-06-29

**Authors:** Dominika Ochnik, Aleksandra M. Rogowska, Cezary Kuśnierz, Monika Jakubiak, Astrid Schütz, Marco J. Held, Ana Arzenšek, Joy Benatov, Rony Berger, Elena V. Korchagina, Iuliia Pavlova, Ivana Blažková, Zdeňka Konečná, Imran Aslan, Orhan Çınar, Yonni Angel Cuero-Acosta, Magdalena Wierzbik-Strońska

**Affiliations:** 1Faculty of Medicine, University of Technology, 40-555 Katowice, Poland; rektorat@wst.com.pl; 2Institute of Psychology, University of Opole, 45-052 Opole, Poland; arogowska@uni.opole.pl; 3Faculty of Physical Education and Physiotherapy, Opole University of Technology, 45-758 Opole, Poland; c.kusnierz@po.edu.pl; 4Faculty of Economics, Maria Curie-Sklodowska University in Lublin, 20-031 Lublin, Poland; monika.jakubiak@poczta.umcs.lublin.pl; 5Department of Psychology, University of Bamberg, 96047 Bamberg, Germany; astrid.schuetz@uni-bamberg.de (A.S.); marco.held@uni-bamberg.de (M.J.H.); 6Faculty of Management, University of Primorska, 6101 Koper, Slovenia; ana.arzensek@fm-kp.si; 7Department of Special Education, University of Haifa, Haifa 3498838, Israel; jbentov2@gmail.com; 8The Center for Compassionate Mindful Education, Tel Aviv 69106, Israel; bergerrony@gmail.com; 9Bob Shapell School of Social Work, Tel-Aviv University, Tel Aviv 69978, Israel; 10Institute of Industrial Management, Economics and Trade, Peter the Great St. Petersburg Polytechnic University, 195251 St. Petersburg, Russia; elena.korchagina@mail.ru; 11Department of Theory and Methods of Physical Culture, Lviv State University of Physical Culture, 79007 Lviv, Ukraine; pavlova.j.o@gmail.com; 12Department of Regional and Business Economics, Mendel University in Brno, 613 00 Brno, Czech Republic; blazkova@mendelu.cz; 13Faculty of Business and Management, Brno University of Technology, 612 00 Brno, Czech Republic; konecna@fbm.vutbr.cz; 14Health Management Department, Bingöl University, Bingöl 12000, Turkey; imranaslan@gmail.com; 15Faculty of Economics and Administrative Sciences, Ataturk University, Erzurum 25240, Turkey; orhanar@gmail.com; 16Faculty of Economics and Administrative Sciences, Ağrı İbrahim Çeçen University, Ağrı 04000, Turkey; 17School of Management, Universidad del Rosario, Bogotá 111711, Colombia; yonnia.cuero@urosario.edu.co

**Keywords:** mental health, anxiety, depression, students, COVID-19, general self-reported health, physical activity, gender, cross-national study

## Abstract

The mental health of young adults, particularly students, is at high risk during the COVID-19 pandemic. The purpose of this study was to examine differences in mental health between university students in nine countries during the pandemic. The study encompassed 2349 university students (69% female) from Colombia, the Czech Republic (Czechia), Germany, Israel, Poland, Russia, Slovenia, Turkey, and Ukraine. Participants underwent the following tests: Patient Health Questionnaire (PHQ-8), Generalized Anxiety Disorder (GAD-7), Exposure to COVID-19 (EC-19), Perceived Impact of Coronavirus (PIC) on students’ well-being, Physical Activity (PA), and General Self-Reported Health (GSRH). The one-way ANOVA showed significant differences between countries. The highest depression and anxiety risk occurred in Turkey, the lowest depression in the Czech Republic and the lowest anxiety in Germany. The χ^2^ independence test showed that EC-19, PIC, and GSRH were associated with anxiety and depression in most of the countries, whereas PA was associated in less than half of the countries. Logistic regression showed distinct risk factors for each country. Gender and EC-19 were the most frequent predictors of depression and anxiety across the countries. The role of gender and PA for depression and anxiety is not universal and depends on cross-cultural differences. Students’ mental health should be addressed from a cross-cultural perspective.

## 1. Introduction

The newly-emerged coronavirus is responsible for a highly viral and infectious disease resulting in a severe acute respiratory syndrome (SARS-Cov-2). The pandemic began in December 2019 and has subsequently spread rapidly worldwide [1].

The first wave of the coronavirus disease (COVID-19) pandemic entered a global stage in the spring of 2020 and prevailed until the summer [2]. The COVID-19 pandemic has forced the introduction of preventive restrictions. Due to the restrictions, social isolation was experienced on an unprecedented scale globally. This contributed to the deterioration of mental health [3,4,5,6,7]. The COVID-19 pandemic is also perceived as the deepest global economic recession in the past eight decades [8]. Considering increased levels of anxiety and depression during previous economic crises [9], the financial instability caused by the pandemic can become a crucial risk factor in relation to mental health deterioration. Research has also shown a linkage between lower social status and mental health issues [10,11]. Therefore, due to financial instability, the current pandemic can affect the mental health of individuals who are not at a serious risk of becoming infected with COVID-19. Recent cross-national studies revealed that mental health deterioration associated with the pandemic is not exclusively limited to individuals who have been infected but extends to the general population [12].

Young adults are highly vulnerable to mental health deterioration during the COVID-19 pandemic [13,14,15] even though they are the least susceptible to the COVID-19 infection [16]. Young age is one of the key risk factors as the prevalence of depressive symptoms in early adulthood is high and dynamic and mediated by several environmental and biological factors [17]. Mental health issues are common in the student population—more than one-third of students experienced some form of mental health problem in the pre-pandemic period [18]. Despite the fact that students are a socially privileged population, they have been at a higher depression risk compared to the general population, even in the pre-pandemic period [19,20]. Students’ physical health status is also relatively poor when compared to their non-studying working peers or the overall population [21,22]. Based on the meta-analysis of studies conducted between 1990 and 2010, the prevalence of depression among students amounted to 30.6% on average [19] compared to 12.9% in the global population based on data from 30 countries collected between 1994 and 2014 [20]. Financial difficulties constitute a risk factor for the increase of anxiety and depression levels. They can also lead to poor academic performance [23]. Financial concerns are not the only factor affecting students’ mental health issues. They can also be influenced by [24] academic pressure and demanding workloads [25], student mistreatment and abuse [26] and worries about health [27]. Students are particularly susceptible to affective disorders due to high social expectations as they are deemed to represent the future of a community [28]. Research showed that during the ongoing pandemic, student status (particularly being a student on the first-cycle of studies) is a relevant risk factor for mental health issues [29,30,31,32]. Social isolation during the COVID-19 pandemic revealed a higher experience of insecurity concerning housing and employment opportunities [33], smaller living space and lower levels of social interaction in young adults compared to adults [4,34]. Academic stress and virtual learning are also crucial risk factors [35,36]. According to the International Labor Organization [37], the education sector has been strongly affected by the COVID-19 pandemic. Therefore, the student population is at a high risk of mental health deterioration during the COVID-19 pandemic and special attention should be paid to research encompassing this cohort.

There are several additional risk factors for mental health deterioration during the ongoing pandemic, such as female gender and lower income [12,31,32,38,39,40,41], place of residence [42,43], financial and learning-related concerns [44,45] or physical inactivity [39]. Concerns regarding loved ones, own health, or academic performance were pronounced during the pandemic [45] and contributed to an increase in anxiety and depression levels [44]. Students also shifted their main concerns from learning-related to financial and/or health-related matters [46]. Recent studies showed that exposure to COVID-19-related matters may increase the risk of anxiety symptoms in students (particularly among men) [47]. Physical activity constitutes the next key predictor of mental health problems. People who spent more time outside during mobility restrictions reported lower stress and higher positive mental health [48]. International research showed that social isolation during the COVID-19 pandemic was linked to lower PA intensity. Additionally, eating patterns were less healthy [49]. Students who were physically inactive (less than 150 min of activity a week) during the COVID-19 pandemic reported higher anxiety and depression compared to the physically active group [39]. Physical activity turned out to be a stronger predictor of depression than anxiety in students [39]. An additional issue related to reactions to the pandemic is mixed media coverage and rapid changes in official messages regarding protective behaviors. Misinformation is one of the crucial factors in anxiety response during the pandemic [50]. Regular searching for additional information concerning the coronavirus turned out to be a risk factor related to the fear of the coronavirus [51].

The number of research papers dedicated to the COVID-19 pandemic has already exceeded the number of studies dedicated to Ebola and H1N1. However, few studies were created via international collaboration [52]. Additionally, cross-national research regarding mental health during the COVID-19 pandemic frequently refers to the general population [12,29,30,31,53,54,55,56,57] rather than the student population [45,58,59]. Additionally, in articles related to students’ mental health, a binational, rather than cross-national perspective appears more frequently [45,58,59]. Cross-national studies concerning mental health during the COVID-19 pandemic indicate that mental health differentiates the general population at a country level [12,29,30,31,53,54,55,56,57]. Analyses from 78 countries showed a slightly higher depression in Poland compared to the overall mean and an even stronger effect in Turkey [30]. The Polish general population manifested the highest anxiety and depression rate during the COVID-19 pandemic compared to the sample from China, Spain, Iran, United States of America, Pakistan, and Vietnam [57].

The main aim of this study is to compare depression and anxiety levels among university students in nine countries: Colombia, the Czech Republic (Czechia), Germany, Israel, Poland, Russia, Slovenia, Turkey, and Ukraine during the first wave of the COVID-19 pandemic. Risk factors for depression and anxiety will also be examined separately in each country, including gender, place of residence, level of study, exposure to COVID-19, the perceived impact of COVID-19 on students’ well-being (including qualifications, economic status, and social relationships), physical activity and physical health.

## 2. Materials and Methods

### 2.1. Participants

The required sample size for each country group was computed a priori by means of G*Power software (Düsseldorf, Germany) [60]. In order to obtain a medium effect size of Cohen’s *W* = 0.03 with given 95% power in a 2 × 2 χ^2^ contingency table, *df* = 1 (two groups in two categories each, two tailed), α = 0.05, G*Power suggests 145 participants are required in each country group (non-centrality parameter λ = 13.05, critical χ^2^ = 3.84, power = 0.95). Initially, the total sample consisted of 2453 respondents. However, 104 persons (4.24% of the initial total sample) declined participation (responded “No” to the informed consent). Therefore, the final total sample encompassed 2349 university students from nine countries: Colombia (*n* = 155), Czechia (*n* = 310), Germany (*n* = 270), Israel (*n* = 199), Poland (*n* = 301), Russia (*n* = 285), Slovenia (*n* = 209), Turkey (*n* = 310), and Ukraine (*n* = 310). The present number of university students in each country exceed the required sample size. This may lead to an increase in the power of 0.95 for the statistical analysis. All the respondents were eligible for the study and confirmed their student status. Additionally, respondents who decided not to reveal their gender were excluded from statistical analyses concerning gender (*n* = 6).

Colombian students (*n* = 155) were recruited from Bogota universities: Del Rosario University (*n* = 142, 92%) and El Bosque University (*n* = 13, 8%). The total sample in Czechia was comprised of students recruited from Mendel University in Brno (*n* = 310, 100%), and in Germany from the University of Bamberg (*n* = 270, 100%). The Israeli sample represented the University of Haifa (*n* = 199, 100%). The Polish sample consisted of 301 students recruited from Maria Curie-Sklodowska University (UMCS) in eastern Poland (*n* = 149, 48%) and from the University of Opole in the south of Poland (*n* = 152, 51%). Russian students were recruited from universities located in Sankt Petersburg: Peter the Great St. Petersburg Polytechnic University (*n* = 155, 54%), Higher School of Economics (HSE) University (*n* = 90, 31%), and St. Petersburg State University of Economics and Finance (*n* = 42, 15%). The total sample in Slovenia was comprised of students recruited from the University of Primorska in Koper (*n* = 209, 100%). Turkish students were from eleven Turkish universities mostly located in eastern Turkey: Bingol University, Bingöl (*n* = 148, 48%); Atatürk University, Erzurum (*n* = 110, 35%); Muğla Sıtkı Koçman University, Muğla (*n* = 35, 11%); Ağrı İbrahim Çeçen University, Ağrı (*n* = 6, 2%); Fırat University, Elazığ (*n* = 3, 0.8%); Kırıkkale University, Kırıkkale (*n* = 1, 0.3%); Adnan Menderes University, Aydın (*n* = 1, 0.3%); Başkent University, (*n* = 3, 1%); Boğaziçi University (*n* = 1, 0.3%), Dicle University, Diyarbakır (*n* = 1, 0.3%), and Istanbul University (*n* = 1, 0.3%). Ukrainian students represented the Lviv State University of Physical Culture (*n* = 310, 100%).

Female students constituted 69% of the sample (*n* = 1627). Over half of respondents lived in rural areas and small towns (*n* = 1248; 54%). First cycle studies (Bachelor) were represented by the highest number of students (*n* = 1843; 78%) compared to the second cycle or higher (*n* = 506; 22%). The majority of participants studied in the full-time mode (*n* = 2007; 85%). The mean age of participants was 23 (*SD* = 4.66). The mean values for depression and anxiety in the total sample were 7.16 (*SD* = 5.52) and 8.85 (*SD* = 6.05), respectively. Detailed descriptive statistics for each country are presented in Appendix A.

All questions included in the Google Forms questionnaire were designated as mandatory. Therefore, participants were unable to omit any response. However, hot-deck imputation was introduced to deal with a low number of missing data (*n* = 5, 0.02%) in the German sample (study conducted via SoSci Survey [61]).

### 2.2. Study Design

The cross-national study was conducted during the first wave of the COVID-19 pandemic (May–July 2020). The sample consisted of 2349 university students from Colombia, Czechia, Germany, Israel, Poland, Russia, Slovenia, Turkey, and Ukraine. The survey study was conducted via Google Forms in all countries except Germany. This country exploited the SoSci Survey [61]. The invitation to participate in the survey was sent to students by researchers via a variety of means, e.g., Moodle e-learning platform, student offices, email, or social media. The average time of data collection was 23.26 min (*SD* = 44.03). No form of compensation was offered as an incentive to participate in eight countries. In Germany, students were offered a possibility to enter into a lottery for a €20 Amazon gift card as an incentive to participate. In Israel, the participants were eligible to win NIS 300 gift cards. To minimize bias sources, the student sample was highly diversified as regards its key characteristics: the type of university, field of study and the cycle of study. Sampling was purposive. The selection criterion was university student status.

Ethics Statement: The study protocol was approved by the ethics committee of the University Research Committee at the University of Opole, Poland, decision no. 1/2020. The study followed the ethical requirements of the anonymity and voluntariness of participation. Each person answered the informed consent question. Following the Helsinki Declaration, a written informed consent was obtained from each student before inclusion.

### 2.3. Measurements

The Patient Health Questionnaire (PHQ-8) [62] was used to measure depression symptoms. The PHQ-8 consists of eight items, conforming with DSM-V diagnostic criteria [48]. The symptoms include depressed mood, loss of interest in most or all activities, loss of energy, or feeling of worthlessness [62]. Participants use a Likert-type response scale ranging from 0 = not at all, to 3 = nearly every day. The range of PHQ-8 scores is from 0 to 24, severe. A cut-off score of 10 or above is recommended to screen for major depressive disorder risk [62]. Due to the requirements of a further statistical analysis with the use of the χ^2^ independence test and logistic regression, the PHQ-8 was dichotomized as follows: 0 = No risk (PHQ-8 < 10), 1 = Risk (PHQ-8 ≥ 10). The internal consistency reliability of the original version measured by Cronbach’s α equals 0.86. The value of 0.88 for the total sample was recorded in this study.

In order to measure anxiety risk, the 7-item Generalized Anxiety Disorder (GAD-7) scale [63] was exploited. GAD-7 is a self-reported measure designed to screen for symptoms following Diagnostic and Statistical Manual of Mental Disorders, fifth edition (DSM-V) criteria [64]. The Generalized Anxiety Disorder (GAD) is characterized by a persistent and excessive worry about various issues. It relates to anxiety as a state [63]. People rate how often they experienced anxiety symptoms in the course of two weeks preceding the study on a 4-point Likert scale (0 = not at all, 1 = several days, 2 = more than half the days, and 3 = nearly every day). The GAD-7 ranges from 0 to 21. Scores above 10 points indicate an anxiety disorder risk [63]. Due to the requirements of the χ^2^ independence test and logistic regression, GAD-7 was dichotomized as follows: 0 = No risk (GAD-7 < 10), 1 = Risk (GAD-7 ≥ 10). The Cronbach’s α for the GAD-7 in this study was 0.92 in the total sample.

Exposure to COVID-19 [39] was assessed based on eight questions regarding the coronavirus consequences: (1) Have you experienced symptoms that could indicate the coronavirus infection?; (2) Have you been tested for the coronavirus?; (3) Were you hospitalized for the coronavirus?; (4) Did you have to be in strict quarantine for at least 14 days, in isolation from loved ones because of the coronavirus infection?; (5) Has anyone in your family, among friends, or relatives been infected with the coronavirus?; (6) Have any of your relatives died of the coronavirus?; (7) Have you or a loved one lost their job because of the coronavirus?; and (8) Are you currently experiencing a worsening of your functioning or economic status due to the coronavirus pandemic’s effects? Individuals answered each of these questions (0 = No, 1 = Yes) The total score was a sum of eight items, where a higher score indicated stronger coronavirus exposure. The results were divided into two categories for the χ^2^ independence test and logistic regression: 0 = Low exposure (score 0), 1 = High exposure (scores 1–8).

The Perceived Impact of Coronavirus (PIC) on students’ well-being [39] was measured using five statements Participants used a 5-item Likert scale (from 1 = I strongly disagree, to 5 = I definitely agree) to express how much they are afraid that the current situation associated with the coronavirus pandemic (COVID-19) may negatively affect their lives in each of the following five aspects: (1) Completing the semester and obtaining qualifications; (2) Finding a job and professional development; (3) Financial situation (e.g., subsistence during studies); (4) Relationships with loved ones, family, (5) Relations with colleagues, friends. Next, scores obtained from the five items were summarized to a total score of the perceived coronavirus impact on students’ well-being (PCI). The higher the scores, the more significant the coronavirus-related concerns were. We have used the median to dichotomize the total score of the PIC and its three subscales: Qualifications (Graduation), Economic Status, and Social Relationships. The total PIC was coded as follows (for the χ^2^ independence test and logistic regression): 0 = Lower (PIC ≤ 15), 1 = Higher (PIC ≥ 16). We added scores of items PIC1 and PIC2 for the Qualifications scale and then coded as 0 = Low (scores 2–6), 1 = High (scores 7–10). Social Relationships scale consisted of items PIC4 and PIC5 coded as 0 = Low (scores 2–4), 1 = High (scores 5–10). Single item PIC3 concerning Economic Status scale was coded as 0 = Low (scores 1–3), 1 = High (scores 4–5). The Cronbach’s α (indicating the internal reliability of the scale) amounted to 0.71 in the present study (in the total sample).

Physical activity (PA) during the coronavirus-related lockdown was assessed using the following question: “How many days a week did you exercise physically or pursued sports activities at home or away from home, at the university, in clubs, or at the gym, in the last month?” [39]. Participants answered this question on an eight-point scale (from 0 = Not one day to 7 = Seven days a week). Next, the students responded to the question: “How many minutes a day (on average) did you practice?” indicating the average number of minutes of PA per day. The number of days was multiplied by the number of minutes per day to calculate the previous week’s PA level. We divided the total sample into two groups: 0 = Sufficient (PA ≥ 150 min weekly) and 1 = Insufficient (PA < 150 min weekly), in line with the WHO recommendation [65].

The General Self-Rated Health (GSRH) status was assessed using two single-item questions as a shorter alternative to the standard general physical health (PH) survey (SF-12V) [66,67]. The first question GSRH-1 concerns an overall physical health (GSRH) assessment (i.e., “In general, would you say your health is…?”), while the second GSRH-2, compares self-health with other people (i.e., “Compared to others your age, would you say your health is…?”) (GSRH Comparative). Both GSRH items are rated on a 5-point Likert scale (1 = Excellent, 2 = Very Good, 3 = Good, 4 = Fair, and 5 = Poor). Therefore, higher scores denote worse health status. Research indicates that poorly self-rated health in the single-item GSRH has a strong association with mortality [66]. We spilt the GSRH as follows (due to the χ^2^ independence test and logistic regression requirements): 0 = Better health (GSRH ≤ 3), 1 = Worse health (GSRH ≥ 4). In the present study, the Cronbach’s α for GSRH was 0.88 (*N* = 2349).

Demographic data included questions regarding age (in years), gender (0 = Men, 1 = Women), place of residence (Village, Town, City, Agglomeration/Metropolis), and the current level of study (Bachelor, Master, Postgraduate, Doctoral). We divided answers regarding the place of residence into two categories (for the χ^2^ independence test and logistic regression) coded as: 0 = village and town, 1 = city, agglomeration, or metropolis. Additionally, we have incorporated 4% of participants who studied at a doctoral or postgraduate level into the category Master. Therefore, for further analysis, the level of study is comprised of two categories: 0 = Bachelor and 1 = Master (for Master or higher).

### 2.4. Statistical Analysis

The statistical analysis included descriptive statistics: mean (M), standard deviation (SD), 95% of confidence interval (CI) with lower limit (LL) and upper limit (UL). Subsequently, a one-way analysis of variance (ANOVA) was performed to test the differences in the mean scores of depression and anxiety between university students from the nine countries: Slovenia, Czechia, Germany, Poland, Ukraine, Russia, Turkey, Israel, and Colombia. The effect size for ANOVA was assessed using η_p_^2^ (a value of η_p_^2^ = 0.01 is considered to be a small effect size, 0.09 a medium effect, and 0.25 a large effect). Tukey’s honest significant difference (HSD) test was used to find means that are significantly different from each other. Furthermore, Pearson’s χ^2^ independence test was conducted to examine relationships between depression and anxiety and other variables in each of the nine countries. A 2 × 2 contingency table was provided in each country separately, for depression and anxiety as independent variables, as well as such predictor variables as gender, place of residence, level of study, physical activity, exposure to the COVID-19 pandemic, the total impact of COVID-19 on students’ well-being, as well as impact in the domain of qualifications, economic status, and social relationships, self-rated physical health, and comparative self-rated physical health (Comparative PH). However, all Colombian students (100%) were assigned to the Town/City category, and 97% (*n* = 155) to the first cycle study. Therefore, place of residence and level of study were excluded from the statistical analysis in the Colombian sample. The effect size for Pearson’s χ^2^ independence test was assessed using φ statistic (a value of φ = 0.1 is considered to be a small effect, 0.3 a medium effect, and 0.5 a large effect). Next, multivariate logistic regression analysis was performed in each country separately to test the adjusted odds ratio (AOR), in order to assess potential risk factors (gender, place of residence, exposure to COVID-19, PIC, PA, PH, Comparative PH) as predictors of depression and anxiety in each country. All predictors were entered into the model simultaneously. The following statistics were calculated for estimation: coefficient estimates, 95% confidence intervals (CI) for the regression coefficient, standard errors of the regression coefficient, odds ratio, *z*-values, and their corresponding *p*-values. The bias-corrected accelerated bootstrapping (BCa) method of estimating regression coefficient was also applied, with the number of replications set to 5000 (if the bias-corrected 95% confidence intervals (CI_B_) did not include the null value, then a statistically significant effect was considered). Goodness of fit of the regression model was assessed using pseudo *R*^2^, including Cox and Snell *R*^2^_CS_, McFadden *R*^2^_McF_, and Nagelkerke *R*^2^_N_. All analyses were performed using Statistica Version 13.1, StatSoft Polska (Cracow, Poland) [68] and the open-source statistical software JASP Version 014.1 [69].

## 3. Results

### 3.1. Country Differences in Depression and Anxiety

A one-way between subjects ANOVA was conducted to compare the effect of country on depression and anxiety (see Figure 1 and Figure 2, for more details). There was a significant effect of country on depression with a large effect size, *F*(8, 2340) = 31.02, *p* < 0.001, η_p_^2^ = 0.09. Post hoc comparisons using the Tukey HSD test indicated that the mean score of the PHQ-8 in Poland was significantly higher than in Slovenia (*p* < 0.01), Czechia (*p* < 0.001), Ukraine (*p* < 0.001), and Germany (*p* < 0.05), and was significantly lower than in Turkey (*p* < 0.001). In addition, Slovenia demonstrated higher depression than Czechia (*p* < 0.01), but lower than Turkey (*p* < 0.001) and Colombia (*p* < 0.05). Among all of the nine countries, the lowest scores in depression emerged in Czechia, where it was significantly lower than that of Russia (*p* < 0.001), Germany (*p* < 0.001), Turkey (*p* < 0.001), Israel (*p* < 0.001), and Colombia (*p* < 0.001). Depression in Ukraine was found as being significantly lower than in Russia (*p* < 0.05), Turkey (*p* < 0.001), and Colombia (*p* < 0.001). Turkey scored the highest in depression, when compared to other countries, including Russia (*p* < 0.001), Germany (*p* < 0.001), Israel (*p* < 0.001), and Colombia (*p* < 0.01). The mean scores for depression are shown in Figure 1.

A significant effect of country on anxiety was also revealed, with a large effect size, *F*(8, 2340) = 57.78, *p* < 0.001, η_p_^2^ = 0.15. As Tukey’s HSD test indicates, the mean score of the GAD-7 in Poland was significantly higher than in Slovenia (*p* < 0.01), Czechia (*p* < 0.001), Ukraine (*p* < 0.001), Russia (*p* < 0.01), and Germany (*p* < 0.001). Slovenia showed significantly higher scores in anxiety than Czechia (*p* < 0.001) and Germany (*p* < 0.001), and lower than Turkey (*p* < 0.001). Czechia demonstrated significantly lower anxiety than Ukraine (*p* < 0.05), Russia (*p* < 0.001), Turkey (*p* < 0.001), Israel (*p* < 0.001), and Colombia (*p* < 0.001), and significantly higher anxiety than Germany (*p* < 0.001). Anxiety level in Ukraine was found as being significantly lower than in Russia (*p* < 0.05), Turkey (*p* < 0.001), Israel (*p* < 0.01), and Colombia (*p* < 0.001), and higher than in Germany (*p* < 0.01). As far as anxiety is concerned, Russia significantly differed from Germany (*p* < 0.001) and Turkey (*p* < 0.01). Among the nine countries, Germany showed the lowest scores in anxiety, significantly lower than Turkey (*p* < 0.001), Israel (*p* < 0.001), and Colombia (*p* < 0.001). In contrast, Turkey demonstrated the highest anxiety, significantly higher than Israel (*p* < 0.001), and Colombia (*p* < 0.01). The mean scores of anxiety are shown in Figure 2.

### 3.2. Association of Depression and Anxiety with Other Variables

A Pearson’s χ^2^ test of independence was performed separately for each country to examine 2 × 2 association between mental health indices, such as depression and anxiety, and such variables as gender, place of residence, level of study, exposure to COVID-19, the perceived impact of COVID-19 on students’ well-being (PIC), including qualifications (graduation), economic status, and relationships, physical activity, and physical health as rated independently and compared with people of the same age. As shown in Appendix A, numerous associations were found for the nine countries for depression and anxiety, respectively.

The relationship between depression and gender was significant in Colombia (χ^2^(1) = 4.44, *p* < 0.05, ϕ = 0.17), Poland (χ^2^(1) = 7.28, *p* < 0.01, ϕ = 0.16), Russia (χ^2^(1) = 10.24, *p* < 0.01, ϕ = 0.19), Turkey (χ^2^(1) = 15.40, *p* < 0.001, ϕ = 0.22), and Ukraine (χ^2^(1) = 9.02, *p* < 0.01, ϕ = 0.17). Place of residence was not significantly associated with depression at all in any country. The level of study was significantly related to depression in Colombia (χ^2^(1) = 4.16, *p* < 0.05, ϕ = −0.16), Czechia (χ^2^(1) = 5.71, *p* < 0.05, ϕ = −0.13), and Slovenia (χ^2^(1) = 8.24, *p* < 0.01, ϕ = −0.19), while it was of no importance in the other countries. The relationship between depression and exposure to COVID-19 was significant in most countries (except Turkey and Colombia), including Slovenia (χ^2^(1) = 0.23, *p* < 0.001, ϕ = 0.33), Czechia (χ^2^(1) = 11.44, *p* < 0.001, ϕ = 0.19), Germany (χ^2^(1) = 4.16, *p* < 0.05, ϕ = 0.12), Poland, (χ^2^(1) = 7.60, *p* < 0.01, ϕ = 0.16), Ukraine (χ^2^(1) = 6.65, *p* < 0.01, ϕ = 0.15), Russia (χ^2^(1) = 10.39, *p* < 0.01, ϕ = 0.19), and Israel (χ^2^(1) = 15.99, *p* < 0.001, ϕ = 0.28). Except Colombia, the total PIC was linked to depression in the following: Slovenia (χ^2^(1) = 33.96, *p* < 0.001, ϕ = 0.40), Czechia (χ^2^(1) = 16.10, *p* < 0.001, ϕ = 0.23), Germany (χ^2^(1) = 39.80, *p* < 0.001, ϕ = 0.39), Poland, (χ^2^(1) = 20.85, *p* < 0.001, ϕ = 0.26), Ukraine (χ^2^(1) = 13.00, *p* < 0.001, ϕ = 0.21), Russia (χ^2^(1) = 19.72, *p* < 0.001, ϕ = 0.26), Turkey (χ^2^(1) = 9.26, *p* < 0.01, ϕ = 0.17), and Israel (χ^2^(1) = 35.64, *p* < 0.001, ϕ = 0.42). Qualifications were insignificant in Czechia and Poland. However, they were of concern in Slovenia (χ^2^(1) = 32.67, *p* < 0.001, ϕ = 0.40), Germany (χ^2^(1) = 16.95, *p* < 0.001, ϕ = 0.25), Ukraine (χ^2^(1) = 4.96, *p* < 0.05, ϕ = 0.13), Russia (χ^2^(1) = 15.66, *p* < 0.001, ϕ = 0.23), Turkey χ^2^(1) = 6.09, *p* < 0.05, ϕ = 0.14), Israel (χ^2^(1) = 12.94, *p* < 0.001, ϕ = 0.26), and Colombia (χ^2^(1) = 3.93, *p* < 0.05, ϕ = 0.16). Deterioration of economic status was a source of concern in most countries (except Poland): Slovenia (χ^2^(1) = 10.10, *p* < 0.01, ϕ = 0.22), Czechia (χ^2^(1) = 8.46, *p* < 0.01, ϕ = 0.16), Germany (χ^2^(1) = 14.02, *p* < 0.001, ϕ = 0.23), Ukraine (χ^2^(1) = 4.65, *p* < 0.05, ϕ = 0.12), Russia (χ^2^(1) = 11.25, *p* < 0.001, ϕ = 0.20), Turkey χ^2^(1) = 13.58, *p* < 0.001, ϕ = 0.21), Israel (χ^2^(1) = 13.12, *p* < 0.001, ϕ = 0.26), and Colombia (χ^2^(1) = 5.06, *p* < 0.05, ϕ = 0.18). Relationships with friends and family members were a source of concern in most countries during the COVID-10 pandemic (except Colombia): Slovenia (χ^2^(1) = 13.70, *p* < 0.001, ϕ = 0.27), Czechia (χ^2^(1) = 11.06, *p* < 0.001, ϕ = 0.19), Germany (χ^2^(1) = 9.76, *p* < 0.01, ϕ = 0.19), Poland (χ^2^(1) = 24.57, *p* < 0.05, ϕ = 0.29), Ukraine (χ^2^(1) = 8.36, *p* < 0.01, ϕ = 0.16), Russia (χ^2^(1) = 24.16, *p* < 0.001, ϕ = 0.29), Turkey χ^2^(1) = 4.26, *p* < 0.05, ϕ = 0.12), and Israel (χ^2^(1) = 14.38, *p* < 0.001, ϕ = 0.27). The relationship between high depression and insufficient physical activity (PA less than 150 min. per week) was revealed in Poland (χ^2^(1) = −4.85, *p* < 0.05, ϕ = −0.13), Ukraine (χ^2^(1) = −11.77, *p* < 0.001, ϕ = −0.20), Russia (χ^2^(1) = −7.36, *p* < 0.01, ϕ = −0.16), and Israel (χ^2^(1) = 3.89, *p* < 0.05, ϕ = −0.14). An association between high depression and worse physical health was significant in most countries (except Czechia and Ukraine): Slovenia (χ^2^(1) = 15.16, *p* < 0.001, ϕ = 0.27), Germany (χ^2^(1) = 21.71, *p* < 0.001, ϕ = 0.29), Poland (χ^2^(1) = 13.14, *p* < 0.001, ϕ = 0.21), Russia (χ^2^(1) = 10.00, *p* < 0.01, ϕ = 0.19), Turkey χ^2^(1) = 5.89, *p* < 0.05, ϕ = 0.14), Israel (χ^2^(1) = 4.89, *p* < 0.05, ϕ = 0.16), and Colombia (χ^2^(1) = 5.14, *p* < 0.05, ϕ = 0.18). When students compared self-rated physical health to other people of the same age, the association between depression and comparative health was noted in most countries (except Russia): Slovenia (χ^2^(1) = 20.88, *p* < 0.001, ϕ = 0.32), Czechia (χ^2^(1) = 14.45, *p* < 0.001, ϕ = 0.22), Germany (χ^2^(1) = 7.09, *p* < 0.01, ϕ = 0.16), Poland (χ^2^(1) = 21.64, *p* < 0.001, ϕ = 0.27), Ukraine (χ^2^(1) = 6.96, *p* < 0.01, ϕ = 0.15), Turkey χ^2^(1) = 10.73, *p* < 0.001, ϕ = 0.19), Israel (χ^2^(1) = 5.76, *p* < 0.05, ϕ = 0.17), and Colombia (χ^2^(1) = 6.83, *p* < 0.01, ϕ = 0.21).

Anxiety was related to gender in Israel (χ^2^(1) = 4.87, *p* < 0.05, ϕ = 0.16), Russia (χ^2^(1) = 4.15, *p* < 0.05, ϕ = 0.12), Turkey (χ^2^(1) = 9.15, *p* < 0.01, ϕ = 0.17) and Ukraine (χ^2^(1) = 7.52, *p* < 0.01, ϕ = 0.16). An association between anxiety and place of residence was significant solely in Poland (χ^2^(1) = 7.67, *p* < 0.01, ϕ = 0.16). The relationship between the level of study and anxiety was observed in Czechia (χ^2^(1) = 6.80 *p* < 0.01, ϕ = −0.15) and Slovenia (χ^2^(1) = 5.61, *p* < 0.05, ϕ = −0.16). Exposure to COVID-19 was significantly associated with anxiety in all countries, Slovenia (χ^2^(1) = 13.25, *p* < 0.001, ϕ = 0.25), Czechia (χ^2^(1) = 10.34, *p* < 0.01, ϕ = 0.18), Germany (χ^2^(1) = 8.82, *p* < 0.01, ϕ = 0.18), Poland (χ^2^(1) = 8.97, *p* < 0.01, ϕ = 0.17), Ukraine (χ^2^(1) = 10.03, *p* < 0.01, ϕ = 0.18), Russia (χ^2^(1) = 6.95, *p* < 0.01, ϕ = 0.16), Turkey χ^2^(1) = 7.90, *p* < 0.01, ϕ = 0.16), Israel (χ^2^(1) = 10.28, *p* < 0.01, ϕ = 0.23), and Colombia (χ^2^(1) = 4.40, *p* < 0.05, ϕ = 0.17). The total PIC was significantly related to anxiety in all countries: Slovenia (χ^2^(1) = 29.98, *p* < 0.001, ϕ = 0.38), Czechia (χ^2^(1) = 13.01, *p* < 0.001, ϕ = 0.20), Germany (χ^2^(1) = 9.36, *p* < 0.01, ϕ = 0.18), Poland (χ^2^(1) = 12.74, *p* < 0.001, ϕ = 0.21), Ukraine (χ^2^(1) = 17.48, *p* < 0.001, ϕ = 0.24), Russia (χ^2^(1) = 5.34, *p* < 0.05, ϕ = 0.14), Turkey χ^2^(1) = 11.92, *p* < 0.001, ϕ = 0.20), Israel (χ^2^(1) = 32.27, *p* < 0.001, ϕ = 0.40), and Colombia (χ^2^(1) = 8.14, *p* < 0.01, ϕ = 0.23). Qualifications (graduation) were an important source of concern in most countries (except in Poland): Slovenia (χ^2^(1) = 23.19, *p* < 0.001, ϕ = 0.33), Czechia (χ^2^(1) = 17.80, *p* < 0.001, ϕ = 0.24), Germany (χ^2^(1) = 11.62, *p* < 0.001, ϕ = 0.21), Ukraine (χ^2^(1) = 11.31, *p* < 0.001, ϕ = 0.19), Russia (χ^2^(1) = 7.37, *p* < 0.01), Turkey χ^2^(1) = 8.78, *p* < 0.01, ϕ = 0.27), Israel (χ^2^(1) = 9.81, *p* < 0.01, ϕ = 0.22), and Colombia (χ^2^(1) = 8.06, *p* < 0.01, ϕ = 0.23). Deterioration of economic status was important in Slovenia (χ^2^(1) = 4.96, *p* < 0.05, ϕ = 0.15), Czechia (χ^2^(1) = 15.81, *p* < 0.001, ϕ = 0.23), Germany (χ^2^(1) = 4.27, *p* < 0.05, ϕ = 0.13), Russia (χ^2^(1) = 4.47, *p* < 0.05, ϕ = 0.13), Turkey χ^2^(1) = 15.00, *p* < 0.001, ϕ = 0.22), and Israel (χ^2^(1) = 4.42, *p* < 0.05, ϕ = 0.15). Excluding Germany, social relationships were of importance in Slovenia (χ^2^(1) = 13.39, *p* < 0.001, ϕ = 0.25), Czechia (χ^2^(1) = 4.27, *p* < 0.05, ϕ = 0.12), Poland (χ^2^(1) = 21.95, *p* < 0.001, ϕ = 0.27), Ukraine (χ^2^(1) = 13.05, *p* < 0.001, ϕ = 0.21), Russia (χ^2^(1) = 10.04, *p* < 0.01, ϕ = 0.19), Turkey χ^2^(1) = 4.87, *p* < 0.05, ϕ = 0.13), Israel (χ^2^(1) = 14.66, *p* < 0.001, ϕ = 0.27), and Colombia (χ^2^(1) = 5.62, *p* < 0.01, ϕ = 0.19). The relationship between insufficient level of physical activity and high anxiety was statistically significant in Poland (χ^2^(1) = −7.94, *p* < 0.01, ϕ = 0.16) and Ukraine (χ^2^(1) = −4.80, *p* < 0.05, ϕ = 0.12). Poor physical health was related to high anxiety in most countries (except Czechia and Israel): Slovenia (χ^2^(1) = 6.45, *p* < 0.05, ϕ = 0.18), Germany (χ^2^(1) = 9.38, *p* < 0.01, ϕ = 0.17), Poland (χ^2^(1) = 18.51, *p* < 0.001, ϕ = 0.25), Ukraine (χ^2^(1) = 6.47, *p* < 0.05, ϕ = 0.14), Russia (χ^2^(1) = 4.97, *p* < 0.05, ϕ = 0.13), Turkey χ^2^(1) = 13.29, *p* < 0.001, ϕ = 0.21), and Colombia (χ^2^(1) = 4.24, *p* < 0.05, ϕ = 0.16). Comparative physical health was linked to anxiety in most countries (except Ukraine, Israel): Slovenia (χ^2^(1) = 10.75, *p* < 0.01, ϕ = 0.23), Czechia (χ^2^(1) = 4.64, *p* < 0.05, ϕ = 0.12), Germany (χ^2^(1) = 4.19, *p* < 0.05, ϕ = 0.13), Poland (χ^2^(1) = 18.56, *p* < 0.001, ϕ = 0.25), Russia (χ^2^(1) = 4.50, *p* < 0.05, ϕ = 0.13), Turkey χ^2^(1) = 11.19, *p* < 0.001, ϕ = 0.19), and Colombia (χ^2^(1) = 8.00, *p* < 0.01, ϕ = 0.23).

### 3.3. Predictors of Depression in the Nine Countries

Multivariate logistic regression was performed to explore significant predictors of depression among a set of variables that were previously included in the relationship analysis using Pearson’s χ^2^ test. An estimation was assessed separately for each country: Colombia (Appendix A), Czechia (Appendix A), Germany (Appendix A), Israel (Appendix A), Poland (Appendix A), Russia (Appendix A), Slovenia (Appendix A), Turkey (Appendix A), and Ukraine (Appendix A). All estimates of the multivariate logistic regressions are shown in Figure 3.

The regression model performed for depression among Colombian university students showed a significant effect only for gender; estimate = 0.76, 95% *CI* = 0.004, 1.516, *SE* = 0.39, OR = 2.14, *Z* = 1.97, Wald’s χ^2^(1) = 3.88, *p* < 0.05 (Appendix A). However, the bias corrected accelerated bootstrapping (BCa) method did not confirm gender to be a significant predictor of depression in Colombian students; bootstrap BCa 95% *CI*_B_ = −0.092, 1.539. The regression model was found to be significant, χ^2^(143) = 21.591, *p* < 0.01, *R*^2^_CS_ = 0.13, *R*^2^_McF_ = 0.10, *R*^2^_N_ = 0.18.

For Czech students (Appendix A), logistic regression showed two predictors of depression, namely exposure to COVID-19 (estimate = 0.66, 95% *CI* = 0.052, 1.267, *SE* = 0.31, OR = 1.93, *Z* = 2.13, Wald’s χ^2^(1) = 4.53, *p* < 0.05) and comparative health (estimate = 1.301, 95% *CI* = 0.396, 2.207, *SE* = 0.46, OR = 3.67, *Z* = 2.81, Wald’s χ^2^(1) = 7.93, *p* < 0.01). On the other hand, bootstrap confirmed only comparative self-rated health as a significant predictor of depression among Czech students (BCa 95% *CI*_B_ = 0.119, 2.223). This model of regression was significant, χ^2^(296) = 38.639, *p* < 0.001, *R*^2^_CS_ = 0.12, *R*^2^_McF_ = 0.12, *R*^2^_N_ = 0.18.

Among German students (Appendix A), depression can be predicted by the total perceived impact of the coronavirus on daily life (estimate = 1.33, 95% *CI* = 0.483, 2.170, *SE* = 0.43, OR = 3.77, *Z* = 3.08, Wald’s χ^2^(1) = 9.51, *p* < 0.01) and physical health (estimate = 1.38, 95% *CI* = 0.419, 0.2.345, *SE* = 49, OR = 3.98, *Z* = 2.81, Wald’s χ^2^(1) = 7.91, *p* < 0.01). These findings were confirmed by the bootstrapping method for both variables, total PIC (BCa 95% *CI*_B_ = 0.351, 2.101) and PH (BCa 95% *CI*_B_ = 0.232, 2.395). The regression model showed adequate significance, χ^2^(253) = 60.82, *p* < 0.001, *R*^2^_CS_ = 21, *R*^2^_McF_ = 0.17, *R*^2^_N_ = 0.28.

For Israeli participants (Appendix A), excluding study level and physical health as predictors of depression, χ^2^(189) = 49.79, *p* < 0.001, *R*^2^_CS_ = 0.22, *R*^2^_McF_ = 0.18, *R*^2^_N_ = 0.30, the regression model was found to be significant. Among two significant predictors of depression, namely, exposure to the coronavirus (estimate = 0.86, 95% *CI* = 0.092, 1.633, *SE* = 0.39, OR = 2.37, *Z* = 2.19, Wald’s χ^2^(1) = 4.81, *p* < 0.05) and total PIC (estimate = 1.35, 95% *CI* = 0.302, 2.394, *SE* = 0.53, OR = 3.85, *Z* = 2.53, Wald’s χ^2^(1) = 6.39, *p* < 0.05), only total PIC was confirmed by the bootstrapping procedure (BCa 95% *CI*_B_ = 0.147, 2.524).

When the regression was performed for Polish students (Appendix A), three variables revealed sufficient significance level using both classical and bootstrapping methods, namely gender (estimate = 0.78, 95% *CI* = 0.190, 1.377, *SE* = 0.30, OR = 2.19, *Z* = 2.59, Wald’s χ^2^(1) = 6.70, *p* < 0.01; BCa 95% *CI*_B_ = 0.115, 1.393), total PIC (estimate = 1.00, 95% *CI* = 0.299, 1.700, *SE* = 0.36, OR = 2.72, *Z* = 2.80, Wald’s χ^2^(1) = 7.82, *p* < 0.01; BCa 95% *CI*_B_ = 0.197, 1.653), and comparative PH (estimate = 1.27, 95% *CI* = 0.286, 2.250, *SE* = 0.50, OR = 3.55, *Z* = 2.53, Wald’s χ^2^(1) = 6.41, *p* < 0.05; BCa 95% *CI*_B_ = 0.126, 2.323). The regression model was found to be significant, χ^2^(288) = 62.46, *p* < 0.001, *R*^2^_CS_ = 0.19, *R*^2^_McF_ = 0.15, *R*^2^_N_ = 0.25.

In the Russian sample of university students (Appendix A), the following predictors of depression were found: gender (estimate = 0.93, 95% *CI* = 0.300, 1.560, *SE* = 0.32, OR = 2.53, *Z* = 2.89, Wald’s χ^2^(1) = 8.37, *p* < 0.01), exposure to the coronavirus (estimate = 0.76, 95% *CI* = 0.080, 1.443, *SE* = 0.35, OR = 2.14, *Z* = 2.19, Wald’s χ^2^(1) = 4.80, *p* < 0.05), PIC-qualification (estimate = 0.90, 95% *CI* = 0.175, 1.624, *SE* = 0.37, OR = 2.46, *Z* = 2.43, Wald’s χ^2^(1) = 5.92, *p* < 0.05), PIC-social relationships (estimate = 1.39, 95% *CI* = 0.681, 2.100, *SE* = 0.36, OR = 4.02, *Z* = 3.84, Wald’s χ^2^(1) = 14.76, *p* < 0.001), physical activity (estimate = −0.78, 95% *CI* = −1.375, −0.176, *SE* = 0.31, OR = 0.46, *Z* = −2.54, Wald’s χ^2^(1) = 6.43, *p* < 0.05), and physical health (estimate = 1.39, 95% *CI* = 0.472, 2.304, *SE* = 0.48, OR = 4.01, *Z* = 2.97, Wald’s χ^2^(1) = 8.83, *p* < 0.01). In addition, bootstrap showed a significant effect for gender (BCa 95% *CI*_B_ = 0.209, 1.590), PIC-qualification (BCa 95% *CI*_B_ = 0.102, 1.636), PIC-social relationships (BCa 95% *CI*_B_ = 0.195, 2.157), and PH (BCa 95% _B_ = 0.125, 2.305). The regression model was significant, χ^2^(271) = 69.38, *p* < 0.001, *R*^2^_CS_ = 0.22, *R*^2^_McF_ = 0.18, *R*^2^_N_ = 0.29.

The regression model conducted in the Slovenian sample of students (Appendix A) showed a good fit, χ^2^(197) = 77.13, *p* < 0.001, *R*^2^_CS_ = 0.31, *R*^2^_McF_ = 0.29, *R*^2^_N_ = 0.43. Among variables, exposure to COVID-19 (estimate = 1.14, 95% *CI* = 0.258, 2.021, *SE* = 0.45, OR = 3.125, *Z* = 2.534, Wald’s χ^2^(1) = 6.423, *p* < 0.05), PIC-qualifications (estimate = 1.26, 95% *CI* = 0.337, 2.183, *SE* = 0.47, OR = 3.52, *Z* = 2.68, Wald’s χ^2^(1) = 7.154, *p* < 0.01), and comparative PH (estimate = 1.61, 95% *CI* = 0.402, 2.825, *SE* = 0.62, OR = 5.02, *Z* = 2.61, Wald’s χ^2^(1) = 6.81, *p* < 0.01) were found to be significant predictors of depression. The bootstrapping method confirmed the significance of all three variables, namely exposure to the coronavirus (BCa 95% *CI*_B_ = 0.058, 2.114), PIC-qualifications (BCa 95% *CI*_B_ = 0.155, 2.076), and comparative PH (BCa 95% *CI*_B_ = 0.165, 2.916).

Although three variables were shown as significant predictors of depression among Turkish students (Appendix A), namely gender (estimate = 0.91, 95% *CI* = 0.398, 1.413, *SE* = 0.26, OR = 2.48, *Z* = 3.50, Wald’s χ^2^(1) = 12.24, *p* < 0.001), PIC-social relationships (estimate = 0.61, 95% *CI* = −0.001, 1.213, *SE* = 0.31, OR = 1.83, *Z* = 1.96, Wald’s χ^2^(1) = 3.84, *p* < 0.05), and comparative PH (estimate = 1.37, 95% *CI* = 0.090, 2.583, *SE* = 0.64, OR = 3.81, *Z* = 2.10, Wald’s χ^2^(1) 4.42, *p* < 0.05), only gender was confirmed using the bootstrapping method (BCa 95% *CI*_B_ = 0.350, 1.429). The model’s fit for the Turkish sample was good, χ^2^(294) = 42.17, *p* < 0.001, *R*^2^_CS_ = 0.13, *R*^2^_McF_ = 0.10, *R*^2^_N_ = 0.18. Among Ukrainian participants (Appendix A), depression can be predicted by gender (estimate = 0.80, 95% *CI* = 0.150, 1,456, SE = 0.33, OR = 2.23, *Z* = 2.41, Wald’s χ^2^(1) = 5.81, *p* < 0.05), exposure to COVID-19 (estimate = 0.87, 95% *CI* = 0.060, 1.686, SE = 0.42, OR = 2.39, *Z* = 2.11, Wald’s χ^2^(1) = 4.43, *p* < 0.05), and PA (estimate = −0.85, 95% *CI* = −1.413, −0.288, SE = 0.29, OR = 0.427, *Z* = −2.97, Wald’s χ^2^(1) = 8.79, *p* < 0.01). Gender (BCa 95% *CI*_B_ = 0.070, 1.512) and PA (BCa 95% *CI*_B_ = −1.396, −0.218) were also found to be significant predictors when the bootstrapping method was used. The regression model presented a good fit, χ^2^(298) = 45.65, *p* < 0.001, *R*^2^_CS_ = 0.14, *R*^2^_McF_ = 0.12, *R*^2^_N_ = 0.20.

### 3.4. Predictors of Anxiety in the Nine Countries

Similarly to the previous analyses, the multiple logistic regression was conducted for anxiety to find predictors among demographic and health-related variables among university students in each country. In Colombian students (Appendix A), only comparative PH was found to be a significant predictor of anxiety (estimate = 1.35, 95% *CI* = 0.132, 2.560, *SE* = 0.46, OR = 3.84, *Z* = 2.17, Wald’s χ^2^(1) = 4.72, *p* < 0.05). However, the bootstrapping procedure did not confirm it (BCa 95% *CI*_B_ = −0.266, 2.819). The model’s fit was sufficient, χ^2^(143) = 25.58, *p* < 0.01, *R*^2^_CS_ = 0.15, *R*^2^_McF_ = 0.13, *R*^2^_N_ = 0.21. All estimates of the multivariate logistic regressions are shown in Figure 4.

Among students in Czechia (Appendix A), the following predictors of anxiety were found: study level (estimate = 1.35, 95% *CI* = 0.132, 2.560, *SE* = 0.46, OR = 3.84, *Z* = 2.17, Wald’s χ^2^(1) = 4.72, *p* < 0.05), exposure to the coronavirus (estimate = 1.35, 95% *CI* = (0.132, 2.560), *SE* = 0.46, OR = 3.84, *Z* = 2.17, Wald’s χ^2^(1) = 4.72, *p* < 0.05), PIC-qualifications (estimate = 1.35, 95% *CI* = 0.132, 2.560, *SE* = 0.46, OR = 3.84, *Z* = 2.17, Wald’s χ^2^(1) = 4.72, *p* < 0.05), PIC-economic status, PA (estimate = 1.35, 95% *CI* = 0.132, 2.560, *SE* = 0.46, OR = 3.84, *Z* = 2.17, Wald’s χ^2^(1) = 4.72, *p* < 0.05), and comparative PH (estimate = 1.35, 95% *CI* = 0.132, 2.560, *SE* = 0.46, OR = 3.84, *Z* = 2.17, Wald’s χ^2^(1) = 4.72, *p* < 0.05). However, bootstrap showed a significant effect only for exposure to the coronavirus (BCa 95% *CI*_B_ = 0.024, 1.764) and PIC-social relationships (BCa 95% *CI*_B_ = 0.100, 2.533). The model presents a good fit, χ^2^(296) = 48.69, *p* < 0.001, *R*^2^_CS_ = 0.15, *R*^2^_McF_ = 0.21, *R*^2^_N_ = 0.27.

The model of regression did not find any statistically significant predictors of anxiety in the sample of German university students (Appendix A). Moreover, bootstrap did not show significance. However, the model’s fit was satisfactory with χ^2^(255) = 33.35, *p* < 0.001, *R*^2^_CS_ = 0.12, *R*^2^_McF_ = 0.30, *R*^2^_N_ = 0.35.

In the sample of students from Israel (Appendix A), the following predictors of anxiety were revealed: gender (estimate = 1.03, 95% *CI* = 0.155, 1.901, *SE* = 0.45, OR = 2.80, *Z* = 2.31, Wald’s χ^2^(1) = 5.32, *p* < 0.05; BCa 95% *CI*_B_ = 0.039, 2.007), and total PIC (estimate = 1.96, 95% *CI* = 0.757, 3.172) *SE* = 0.62, OR = 7.13, *Z* = 3.19, Wald’s χ^2^(1) = 10.17, *p* < 0.001; BCa 95% *CI*_B_ = 0.471, 3.300), which was also confirmed using the bootstrap procedure. The level of study and PH were not included in the model because these variables did not sufficiently meet the criteria. The goodness of fit for the regression model was adequate, χ^2^(189) = 48.23, *p* < 0.001, *R*^2^_CS_ = 0.22, *R*^2^_McF_ = 0.19, *R*^2^_N_ = 0.30.

Among Polish participants (Appendix A), four predictors of anxiety were presented in the regression model: place of residence (estimate = −0.66, 95% *CI* = −1.271, −0.039, *SE* = 0.31, OR = 0.52, *Z* = −2.09, Wald’s χ^2^(1) = 4.35, *p* < 0.05), exposure to the coronavirus (estimate = 0.77, 95% *CI* = 0.165, 1.365, *SE* = 0.31, OR = 2.15, *Z* = 2.50, Wald’s χ^2^(1) = 6.24, *p* < 0.05), PIC-social relationships (estimate = 1.24, 95% *CI* = 0.513, 1.970, *SE* = 0.37, OR = 3.46, *Z* = 3.40, Wald’s χ^2^(1) = 11.15, *p* < 0.001), and PH (estimate = 1.98, 95% *CI* = 0.297, 3.689, *SE* = 0.86, OR = 7.23, *Z* = 2.31, Wald’s χ^2^(1) = 5.32, *p* < 0.05). However, only two of these were confirmed by the bootstrapping method: exposure (BCa 95% *CI*_B_ = 0.069, 1.403) and PIC-social relationships (BCa 95% *CI*_B_ = 0.432, 2.004). The model of regression presented a good fit, χ^2^(288) = 64.78, *p* < 0.001, *R*^2^_CS_ = 0.19, *R*^2^_McF_ = 0.16, *R*^2^_N_ = 0.26.

The perceived impact of the coronavirus on social relationships was the sole predictor of anxiety for Russian university students (Appendix A), estimate = 1.15, 95% *CI* = 0.446, 1.843, *SE* = 0.36, OR = 3.14, *Z* = 3.21, Wald’s χ^2^(1) = 10.32, *p* < 0.001; BCa 95% *CI*_B_ = 0.318, 1.973. The regression model showed a sufficient fit, χ^2^(271) = 34.05, *p* < 0.001, *R*^2^_CS_ = 0.11, *R*^2^_McF_ = 0.10, *R*^2^_N_ = 0.16.

Although the model’s fit statistics were appropriate, χ^2^(197) = 53.424, *p* < 0.001, *R*^2^_CS_ = 0.23, *R*^2^_McF_ = 0.22, *R*^2^_N_ = 0.33, the variables included into the regression model were not found to be predictors of anxiety in the sample of Slovenian students (Appendix A).

In the Turkish group of students, gender (estimate = 0.66, 95% *CI* = 0.162, 1.151, *SE* = 0.25, OR = 1.93, *Z* = 2.60, Wald’s χ^2^(1) = 6.77, *p* < 0.01) and exposure to the coronavirus (estimate = 0.68, 95% *CI* = 0.032, 1.319, *SE* = 0.33, OR = 1.97, *Z* = 2.06, Wald’s χ^2^(1) = 4.24, *p* < 0.05) were found to be significant predictors of anxiety (Appendix A). However, only gender was confirmed by the bootstrapping test (BCa 95% *CI*_B_ = 0.111, 1.175). The regression model’s fit was adequate, χ^2^(294) = 43.17, *p* < 0.001, *R*^2^_CS_ = 0.13, *R*^2^_McF_ = 0.10, *R*^2^_N_ = 0.18.

Both gender (estimate = 0.88, 95% *CI* = 0.148, 1.603, *SE* = 0.37, OR = 2.40, *Z* = 2.36, Wald’s χ^2^(1) = 5.57, *p* < 0.05) and PIC-social relationships (estimate = 0.82, 95% *CI* = 037, 1.593, *SE* = 0.40, OR = 2.26, *Z* = 2.05, Wald’s χ^2^(1) = 4.21, *p* < 0.05) were shown to be significant predictors of anxiety in the sample of Ukrainian university students (Appendix A). However, when bootstrap was performed, only gender was a predictor of anxiety (BCa 95% *CI*_B_ = 0.101, 1.639).

## 4. Discussion

Our study showed the importance of a cross-national perspective in exploring university students’ mental health during the first wave of the COVID-19 pandemic. The research revealed large differences in depression and anxiety rates in the nine countries. The highest rates of depression and anxiety were noted in Turkish students, whereas the lowest depression was reported in Czech students and the lowest anxiety in German students. We have also shown the associations of mental health with other variables in the nine countries. Physical activity turned out to be associated with mental health in less than half of the countries. However, the following factors proved to be associated with anxiety and depression in most of the countries: exposure to COVID-19, the perceived impact of COVID-19 on students’ well-being, including graduation, economic status, and relationships quality, and general and comparative health.

The study revealed a variety of depression and anxiety risk factors in prediction models for each country. Even though certain variables, such as exposure to COVID-19, were significant among students across the nine countries, other key variables, e.g., gender or physical activity, were credible only in particular countries. Therefore, our results underline the necessity for the cultural context to be taken into account when exploring mental health in the student population.

### 4.1. Anxiety and Depression across the Nine Countries

Cross-national analyses allow for the obtained results to be analyzed from the global perspective. Previous cross-national research in European countries showed that a lower risk of depression is associated with socioeconomic factors [70]. In addition, a study using the Human Development Index (HDI) revealed that the highest depression prevalence was in medium HDI countries [20]. People in medium HDI countries might be subjected to more stressors compared to low HDI countries due to high expectations accompanied with high living costs and the cost of depression treatment [71]. However, in our research, all countries, except Ukraine and Colombia, are designated as very high HDI countries (HDI ≥ 0.800). Despite the above, large differences concerning depression in relation to the individual country were revealed. This suggests that other circumstances should be taken into account to explain the results of depression at a cross-national level. In certain cases, our results were in contrast to expectations, since e.g., Turkish students (very high HDI) exceeded all other students in the depression level, whereas Ukrainian students (high HDI) scored similarly to Czech students (very high HDI), thus presented the lowest level of depression in this study. Additionally, students from Czechia, which occupies 27th position in the HDI ranking, scored significantly lower in depression compared to students in Germany holding 6th position in this global ranking. Those inconsistent results may arise from the fact that the student population differs significantly from the general population. Moreover, the estimation of HDI change for 2020 globally shows an unprecedented shock to human development as all of HDI capabilities (long and healthy life, being knowledgeable, and having a decent standard of living) were severely affected indicating unparalleled acute decrease even compared to the post-2007 global financial crisis [72].

On the other hand, Turkish students, who reported the highest depressive symptoms among the nine countries, mostly live in eastern Turkey, where living standards are significantly lower compared to western Turkey [73]. The highest depression and anxiety levels in Turkey compared to the other nine countries may be explained by other significant factors. For example, in Turkey, the pandemic situation may affect student lifestyle by overlapping with other socioeconomic burdens, the current volatile economic situation [74], and high unemployment among young people, reaching 30% [75].

Nonetheless, when analyzing depression and anxiety levels in the nine countries from the global socioeconomic perspective, it appears that a country’s financial situation and HDI may play a more prominent role in anxiety than depression. Students from Germany—the country with the highest HDI and the highest possible score in Standard & Poor’s Global Ratings [76]—presented a minimal anxiety level, which notably, was the lowest among the nine countries. Among other predictors and associated variables, this may also be due to a more stable economy at the national level. Financial security may play a more significant role in decreasing anxiety than depression among students, as the German example shows.

### 4.2. Association of Depression and Anxiety Risk with Other Variables

Exposure to COVID-19, perceived impact of COVID-19 on students’ well-being, including graduation, economic status, and relationships’ quality, as well as general and comparative health, turned out to be associated with anxiety and depression in the majority of the surveyed countries. Therefore, these variables can be designated as key mental health risk factors from the cross-cultural perspective.

Even though previous research has shown the importance of physical activity for anxiety and depression in the student population [39], this study clearly manifests that the cultural context should be taken into account when analyzing this issue. Physical activity was associated with anxiety only in two (Poland and Ukraine) out of nine countries. The association with depression was revealed only in four countries (Poland, Ukraine, Russia and Israel). Therefore, in most countries, physical activity was not associated with mental health in students. One of the factors related to physical activity from the cross-cultural perspective is motivation to participate, as in individualistic cultures, like USA, the key motivation was competition, whereas in collectivistic cultures, like China, it was rather a social affiliation and wellness [77].

Living in a city or town/village turned out to be irrelevant for depression in the nine countries. This is in congruence with previous meta-analyses [20,78,79]. Students are quite a homogeneous group. Therefore, both groups living in rural and urban areas have common characteristics (i.e., young age) linked to depressive symptoms during the pandemic [13,14,15]. However, living in a small city or a village was associated with anxiety risk exclusively in Polish students. In all the remaining countries, it was an irrelevant factor. Previous research also showed inconsistent results. Living in an urban area was linked to lower anxiety in China [42] but higher anxiety in Bangladesh [43].

Our research showed that the association between gender and mental health risk is not clear when analyzed in different countries. Female students from Ukraine, Russia, and Turkey had a higher prevalence of anxiety and depression risk compared to male students in those countries. The highest rate of both depression and anxiety risk was revealed among Turkish female students. Furthermore, Polish and Israeli female students showed a higher prevalence of anxiety risk, whereas Colombian female students manifested the risk of depression. Previous research showed a gender effect on the prevalence of depression [20,70,79,80]. Additionally, a recent meta-analysis regarding students’ mental health during the COVID-19 pandemic has revealed that female students were found to have a higher prevalence of anxiety and depression [81]. However, we have found no gender association with mental health issues among Slovenian, Czech, and German students. Therefore, the cultural context should be incorporated when exploring gender association with mental health issues.

### 4.3. Predictors of Depression and Anxiety in the Nine Countries

The multiple regression models are closer to actual psychological complexity, as they reveal risk factors in their simultaneous effect on mental health, compared to bivariate models where the particular factors predict mental health issues independently. The most frequent predictors of depression and anxiety in the nine countries were gender, exposure to COVID-19, and comparative physical health.

The multiple logistic regression proved gender to be a significant predictor of anxiety but only among Israeli, Ukrainian, and Turkish students. Gender was a more frequent predictor of depression among Colombian, Polish, Russian, Turkish, and Ukrainian students, but a less significant predictor in Colombia. Previous cross-national research in 23 European countries showed that the largest gender differences in depression were noted in certain former Soviet Union countries, and the lowest in Western and Nordic countries [70]. The results in our study partially conform with the aforementioned report. However, in our research, gender was not a risk factor for mental health issues among students in Slovenia and Czechia (former Soviet Union countries). This inconsistency can be partially explained by gender inequalities denoted by the the Gender Inequality Index (GII) [82], which in Slovenia (0.07) and Czechia (0.14) is relatively lower compared to Ukraine (0.29) and Russia (0.25). Therefore, the gender role hypothesis seems to be a more appropriate explanation, particularly for female gender as a risk factor for depression in five out of the nine countries.

The gender role hypothesis claims that the gender gap in the prevalence of mental health issues is due to specific differences in coping resources, stressors, or opportunities for expressing psychological distress distinctively for women and men [83]. Gender role (the concept of femininity and masculinity) affects major risk factors for internalizing and externalizing problems [84]. This hypothesis has found a partial confirmation as regards depression, but not anxiety [83]. Depression was revealed to be related to the changes in traditional female gender roles. Narrowing gender differences in depression was observed along with the declining gender role traditionality [83]. Our study also confirms the significance of gender as a predictor of depression in relation to the gender role hypothesis. However, the results in Israel, Ukraine and Turkey also show the significance of gender in predicting anxiety in the student population during the COVID-19 pandemic.

Multiple regression models showed the importance of exposure to the COVID-19 infection in five countries (Slovenia, Czechia, Israel Russia, and Ukraine) for depression, and in four countries (Czechia, Poland, Turkey, and Ukraine) for anxiety, even though the stringency of restrictions index (ranging from 0 to 100) in those countries varied from 41 in Slovenia to 82 in Ukraine. Therefore, exposure to the infection as a risk factor of depression or/and anxiety appeared in several countries independently of restrictions introduced by the governments.

The perceived impact of COVID-19 on students’ well-being was a risk factor for depression in Israel and Germany, and additionally, for anxiety in Israel. In other countries, this variable was insignificant in multivariate models. However, its subscales showed different patterns depending on the country. Worries about graduation were considered as risk factors for depression in Slovenia and Russia, and for anxiety in Czechia and Russia. The perceived impact of COVID-19 on students’ economic status was significantly associated with depression and anxiety in the majority of the countries, as the above analysis showed. The deterioration of economic status as a risk factor for both depression and anxiety is in line with other studies [85,86]. However, when economic status was introduced in multiple models, it turned out to be a trivial predictor of depression, while being a significant predictor of anxiety only in one country (Czechia). Therefore, even though PIC Economic Status is relevant when analyzed as a singular risk factor for mental health, when combined with other risk factors for mental health, such as exposure to COVID-19 or female gender, it becomes insignificant.

Concern about relationship quality was the strongest predictor in the multivariate models of depression and anxiety in Poland and Russia. Therefore, in the countries with stronger traditional family values, the perceived impact of the COVID-19 pandemic on students’ relationships with family was a significant risk factor for mental health issues. Insufficient physical activity in multivariate models was a risk factor for depression in Russia and Ukraine, and for anxiety in Czechia. Worse physical health played a different role than worse comparative physical health. General physical health was a strong predictor of depression in Germany and Russia and a weaker predictor of anxiety in Poland. Worse comparative health turned out to be a significant risk factor for depression in four countries (Czechia, Poland, Slovenia, and a weaker predictor in Turkey). For anxiety, this was true only for two countries. However, the results in Colombia and Czechia were not confirmed when the bias-corrected accelerated bootstrapping method was introduced. Therefore, comparative physical health was a more common predictor of depression than anxiety among students across the nine countries.

Although introduced variables allowed for the creation of multivariate models of depression in each country, anxiety was not explained by proposed predictors in Slovenia, Germany, and Colombia.

### 4.4. Limitations

There are several limitations to the present study. One is the cross-sectional character of the research. The longitudinal study could reveal the cause–-effect relationship between the proposed indices and mental health issues. Direct comparisons among countries are also limited due to the different pace and extent of public health restrictions imposed by governments and due to the situation with COVID-19 related deaths in the observed period in each of the observed countries. Another limitation is a self-selected study sample and data collection via self-reported questionnaires. Therefore, the data can be subject to retrospective response bias. Previous research showed that more depressive symptoms can be elicited for milder forms of depression through self-reported measurements compared to clinician-rating methods (interview) [87]. More educated and younger people usually score higher on self-rated scales than on clinician-rating scales [88]. However, it should be noted that even though a milder form of depression may be elicited among young adults, depressive symptoms have increased during the pandemic [89,90]. Finally, generalizing the results may be hindered by the lack of random sampling and representation of the student population being limited to specific regions in each country.

Considering strengths and limitations of this study, future research ought to examine mental health using a longitudinal design from the cross-cultural perspective.

## 5. Conclusions

Our study has shown risk factors for depression and anxiety and differences in mental health among university students in the nine countries during the first wave of the COVID-19 pandemic. We have revealed that even so common a risk factor as gender does not predict anxiety or depression in all the countries. Moreover, physical inactivity as a risk factor strongly depends on the country, and in most of the nine countries was a significant predictor neither for anxiety nor depression.

This research underlines the necessity of interpreting data within the cross-cultural context and argues that presenting mental health results during the COVID-19 pandemic only in one country can be challenging in terms of generalization. We demonstrated that, even though there are several risk factors associated with mental health issues in all of the nine countries (i.e., exposure to COVID-19, perceived impact of COVID-19 on students’ well-being, including graduation, economic status, and relationships quality, general and comparative health), the multivariate models differed drastically among the countries. Therefore, despite the globalization of a homogeneous student population, our study showed varied mental health predictors in relation to cultural, political and economic situation in a particular country. Planning and implementation of psychological intervention programs for students should include differentiation by country concerning mental health risk factors.

## Figures and Tables

**Figure 1 jcm-10-02882-f001:**
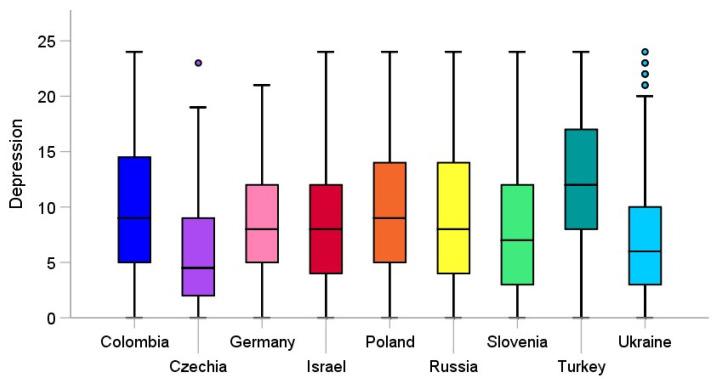
Mean scores of depression (PHQ-8) among university students across the nine countries during the first wave of the COVID-19 pandemic. The dots in the figure represent outliers.

**Figure 2 jcm-10-02882-f002:**
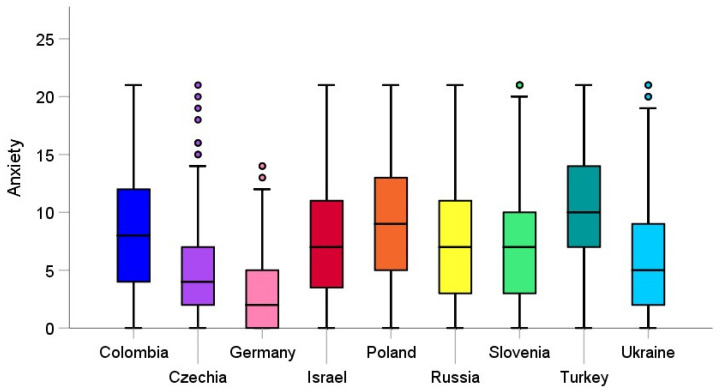
Mean scores for anxiety (GAD-7) among university students across the nine countries during the first wave of the COVID-19 pandemic. The dots in the figure represent outliers.

**Figure 3 jcm-10-02882-f003:**
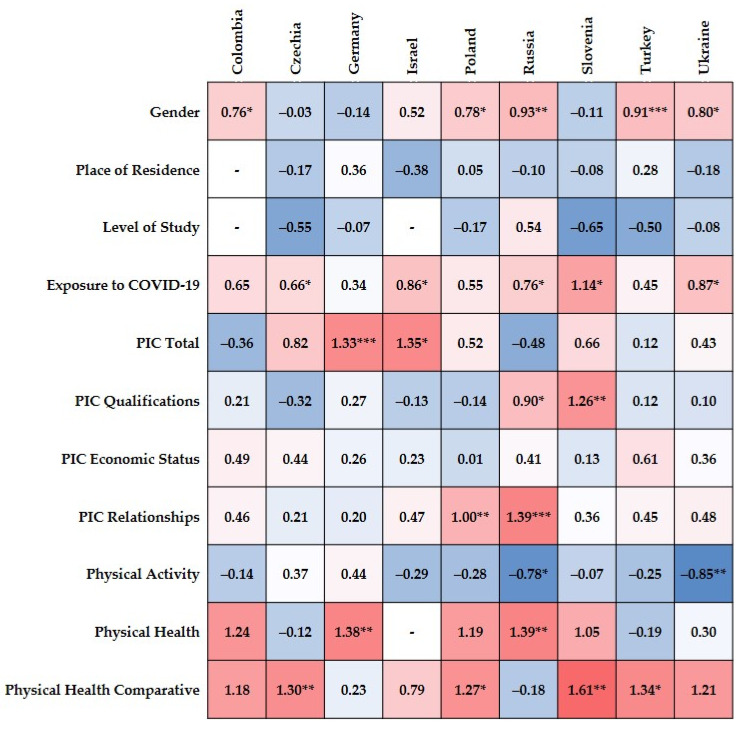
Logistic regression estimates heatmap for depression symptoms among university students from Colombia, Czechia, Germany, Israel, Poland, Russia, Slovenia, Turkey, and Ukraine. Positive estimates are marked in red, negative estimates are marked in blue. PIC = Perceived Impact of COVID-19 on Students’ Well-being. * *p* < 0.05, ** *p* < 0.01, *** *p* < 0.001.

**Figure 4 jcm-10-02882-f004:**
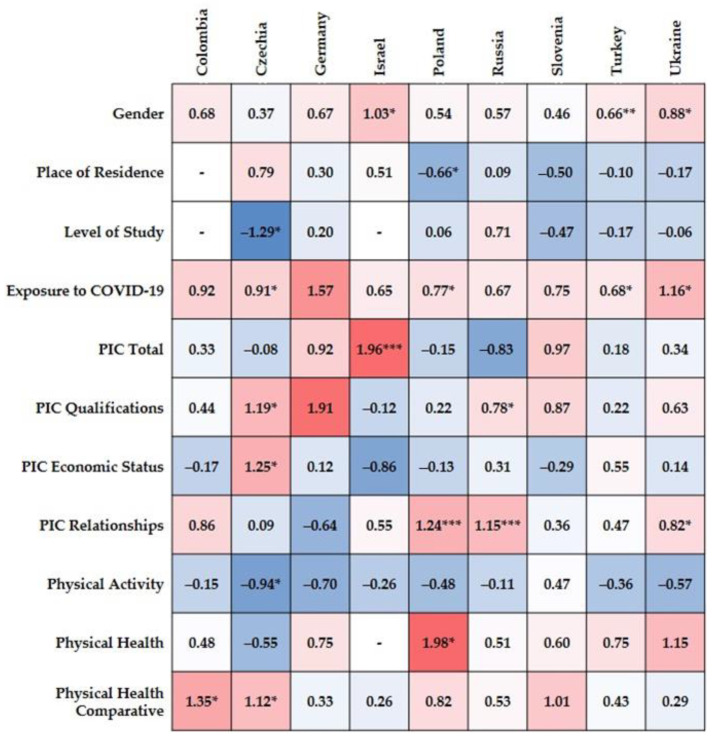
Logistic regression estimates heatmap for anxiety symptoms among university students from Colombia, Czechia, Germany, Israel, Poland, Russia, Slovenia, Turkey, and Ukraine. Positive estimates are marked in red, negative estimates are marked in blue. PIC = Perceived Impact of COVID-19 on Students’ Well-being. * *p* < 0.05, ** *p* < 0.01, *** *p* < 0.001.

## Data Availability

The materials and methods are accessible at the Center for Open Science (OSF), titled: Well-being of undergraduates during the COVID-19 pandemic: International study [91]. The datasets generated during and/or analyzed during the current study are available from the corresponding author on reasonable request.

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
