# Peer review of "A Comparison of Depression and Anxiety among University Students in Nine Countries during the COVID-19 Pandemic"

_jcm, 2021, doi:10.3390/jcm10132882_

Round 1

Reviewer 1 Report

Manuscript: A Comparison of Depression and Anxiety among University 2 Students in Nine Countries during the COVID-19 Pandemic     The present study presents us with a n=2,349 from nine countries: Slovenia, the Czech Republic, Poland, Ukraine, Germany, Russia, Turkey, Israel, and Colombia. Researchers applied a battery of tests related to Health, Anxiety, Depression, Exposure to COVID-19 and Physical Activity showing significant differences between countries, concluding that students’ mental health should be addressed from cross-cultural perspective.   I find this study well written and designed.   I suggest discussing the potential role of the COVID-19 media coverage in either enhancing or mitigating anxiety related to COVID-19.   I suggest the following readings:   Ammar, A., Brach, M., Trabelsi, K., Chtourou, H., Boukhris, O., Masmoudi, L., ... & Hoekelmann, A. (2020). Effects of COVID-19 home confinement on eating behaviour and physical activity: results of the ECLB-COVID19 international online survey. Nutrients, 12(6), 1583.   Coelho, C. M., Suttiwan, P., Arato, N., & Zsido, A. N. (2020). On the nature of fear and anxiety triggered by COVID-19. Frontiers in psychology, 11, 3109.   Mertens, G., Gerritsen, L., Salemink, E., and Engelhard, I. (2020). Fear of the
coronavirus (COVID-19): predictors in an online study conducted in march
2020. J. Anxiety Disord. 74:102258. doi: 10.1016/j.janxdis.2020.102258

Author Response

Response to Editor and Reviewer 1

We want to thank the Reviewer for the highly detailed evaluation of our manuscript and constructive comments. We do appreciate the valuable contribution and believe that it helped to improve our manuscript. We have addressed and incorporated all of the Reviewers’ comments. Our response is marked in italic.

Please see the provided changes in the attached manuscript. 

Response to Reviewer 1

  1. Manuscript: A Comparison of Depression and Anxiety among University 2 Students in Nine Countries during the COVID-19 Pandemic   The present study presents us with a n=2,349 from nine countries: Slovenia, the Czech Republic, Poland, Ukraine, Germany, Russia, Turkey, Israel, and Colombia. Researchers applied a battery of tests related to Health, Anxiety, Depression, Exposure to COVID-19 and Physical Activity showing significant differences between countries, concluding that students’ mental health should be addressed from cross-cultural perspective. I find this study well written and designed.
  • Thank you for this comment.

  1. I suggest discussing the potential role of the COVID-19 media coverage in either enhancing or mitigating anxiety related to COVID-19.   I suggest the following readings:  
  • Ammar, A., Brach, M., Trabelsi, K., Chtourou, H., Boukhris, O., Masmoudi, L., ... & Hoekelmann, A. (2020).Effects of COVID-19 home confinement on eating behaviour and physical activity: results of the ECLB-COVID19 international online survey. Nutrients, 12(6), 1583.  
  • Coelho, C. M., Suttiwan, P., Arato, N., & Zsido, A. N. (2020). On the nature of fear and anxiety triggered by COVID-19. Frontiers in psychology, 11, 3109.  
  • Mertens, G., Gerritsen, L., Salemink, E., and Engelhard, I. (2020). Fear of the coronavirus (COVID-19): predictors in an online study conducted in march 2020. J. Anxiety Disord. 74:102258. doi: 10.1016/j.janxdis.2020.102258
  • Thank you for this suggestion. We have enriched the introduction with the suggested literature:
    • lines 11-112: “International research showed that social isolation during the COVID-19 pandemic was linked to lower PA intensity. Additionally, eating patterns were less healthy (Ammar et al., 2020).
    • lines 115-119: “An additional issue related to reactions to the pandemic is mixed media coverage and rapid changes in official messages regarding protective behaviors. Misinformation is one of the crucial factors in anxiety response during the pandemic (Coelho et al., 2020). Regular searching for additional information regarding the coronavirus turned out to be a risk factor for fear of coronavirus (Mertens et al., 2020).” 

Reviewer 2 Report

The reviewed paper deals with mental health of young adults during the COVID-19 pandemic. In particular, the authors of this paper present a cross-sectional online survey study in which depression and anxiety among University students in nine countries are compared. Cross-national data collection took place during the first wave of the coronavirus disease in the spring of 2020. After presentation of descriptive statistics, the authors used inferential methods (e.g., ANOVA) to find differences between the nine countries. After that, associations between depression/anxiety risks and other variables were analyzed with the two-dimensional chi-squared test for independency. The authors identified a large number of risk factors for depression and anxiety among University students with logistic regression analyses. Interestingly, these results were not homogenous but rather varied across countries. Thus, the authors concluded that planning and implementation of psychological intervention programs for students should include a country differentiation concerning mental health risk factors.

I really enjoyed reading this article and thank the authors for the opportunity to review their interesting paper. Overall, I believe this is a very important paper in the current health crisis. Due to the COVID-19 pandemic, a lot of psychological health issues (e.g., depression, loneliness, anxiety) arose in the human population in the last year. I think it is worth to shed more light on this dramatic situation, not only from the physical health perspective (e.g., fever, dry cough, tiredness) but also from a psychological perspective. In the face of the worldwide COVID-19 pandemic it makes absolutely sense to study and compare depression and anxiety across different countries. I appreciate authors’ efforts in conducting a large cross-national online survey study in nine countries.

However, I identified several issues in the current version of this paper. Some are theoretical in nature others are related to presented statistical analyses. Some are minor others are major. Below I listed some comments/suggestions that I hope will help the authors to further develop this line of work. So, let’s go through this:

  1. General comment: One of my major concerns about this paper are several serious breaks and flaws in writing style. Some parts of the paper were really good and nice to read, other parts were absolutely confusing (e.g., presentation of logistic regression analyses, chapter 3.3, page 9, line 385). I guess, the authors split up the paper and some persons are responsible for writing special parts of the paper (e.g., theoretical background, presentation of empirical results and so on). However, as a reviewer and reader, I felt this missing homogeneity so much. For example, the order of measurements in the abstract (page 1, line 39) did not match the order of measurements in the measurement section (chapter 2.2, page 3, line 145). The presentation of the ANOVA was nearly perfect (chapter 3.1, page 6, line 298), the presentation of chi-squared tests and logistic regression analyses (chapter 3.3., page 9, line 385) was very confusing. Again, I think this paper has big potential, especially the cross national data collection, but large parts of this article must be re-written and re-structured before a publication of this article makes sense.
  2. Theoretical background: A space is missing behind the dot and the word “Despite” („period [18].Despite“, page 2, line 77). Please add a space.
  3. Theoretical background: “Mental health issues are common in the student population— more than one-third of students experienced some form of mental health problems in the pre-pandemic period [18].” (page 2, line 75 to 77). I did not get the point, why University students are so prone for depression and anxiety. The same is true for other populations (e.g., pupils). Why do the authors focus on this special population of University students? They cite a lot of literature in this context but missed to clarify their arguments. It is not sufficient that readers must search for themselves to get an appropriate answer to that question. I can formulate some hypotheses on my own (e.g., living on their own, no or reduced contact to colleagues, no learning groups, no financial income) but also for the opposite (e.g., living by parents, online learning platforms). Hence, I urge the authors to add some passages to the theoretical background that explain and clarify the vulnerability of University students for depression and anxiety.
  4. Theoretical background (page 3, line 118): not “CIOVID” but rather “COVID”
  5. Chapter 2.2 Measurements: As mentioned in my point 1, the order of measurements in the abstract (PHQ-8, GAD-7, […], page 1, line 39) do not match the order of measurements in the measurement section (GAD-7, PHQ-8, […], chapter 2.2, page 3, line 145). Please keep a consisted writing style and correct this, so that the order of measurements is constant throughout the whole manuscript.
  6. Chapter 2.2 Measurements (page 4, line 169): It seems that the instrument “Exposure to COVID-19” is self-constructed by the authors. Please add this hint or, in the other case, cite appropriate literature.
  7. Chapter 2.2 Measurements (page 4, line 182): It seems that the instrument “Perceived Impact of Coronavirus (PIC)” is self-constructed by the authors. Please add this hint or, in the other case, cite appropriate literature.
  8. Chapter 2.2 Measurements (page 5, line 222): Here is a good example of missing consistency throughout the whole manuscript. The authors add behind the Cronbach’s α value the (reduced) sample size (N=2,349). However, at this point the reader can’t know that the authors recruited 2,453 respondents but 104 persons declined participation, resulting in a final sample size of 2,349 (see chapter 2.3 Participants, page 5, line 233). This information is presented in a later chapter so that I claim an infrigment of consistency which probably occurred of fractured writing. Probably, Person A wrote chapter 2.2, Person B wrote chapter 2.3 and no adjustment between the two chapters occurred. Thus, I urge the authors to correct this and keep a consistent writing style.
  9. Chapter 2.3 Participant (page 6, line 266): “The sample size for each country was computed by the means of G*Power software”. As a statistician, I appreciate very much that the authors did a sample size calculation a-priori before the recruitment started. However, they missed to present the actual values (significance level α, assumed power, and effect size) to calculate the minimum number of participants per each country needed. In scientific papers, it is important to replicate the calculated sample size for transparency reasons. How did the authors get the number 159? The authors report the underlying statistical test (chi-squared test), hint to the significance level α (p<0.05 and 95% confidence intervals -> α seems to be 5%), and the two-tailed nature of the test only. However, which statistical power and which effect size did they use for the sample size calculation and why? Is there any rationale for the chosen (but not presented) effect size, any literature that could be used for strengthen the assumed effect size? I urge the authors to re-write this section and report all necessary values.
  10. Chapter 3.1 (page 7, line 329): Here, the authors illustrate measured depression and anxiety across the nine countries with bar charts (Figure 1 and Figure 2). Although this is often done in scientific papers, it is still wrong. Why? Bar charts represent mean values which is one value only, a dot, and not a bar. Hence, bar charts make no sense for illustrating mean values. As a statistician, I would recommend using box plots (in full word: box-whisker plots) instead. Box plots are in this situation more appropriate than bar charts, because they represent not one single value but rather the distribution of measured values. Using box plots has the advantage that the reader gets an idea about the range of values in the sense of transparent distribution. Median, first and third quantile as well as minimum and maximum values, even outliers can be detected within box plots. Hence, I urge the authors to replace the bar charts and use box plots instead for illustrating their measured depression and anxiety values.
  11. Chapter 3.1 (page 7, line 329): Something strange happened to Footnotes of Figure 1 and Figure 2 (smaller line spacing in Figure 2 compared to Figure 1). Please adjust this.
  12. Chapter 3.2 (page 8, line 342): Here is another good example of missing consistency throughout the whole manuscript. An unnecessary repetition occurs in this section. As a statistician, I really appreciate that authors clearly describe their codings (e.g., “coded No Risk = 0”, page 8, line 342). However, they already introduced their codings in chapter 2.2 Measurements (page 3, line 146). I guess, it is the same problem as before: Person A wrote chapter 2.2, Person B wrote chapter 3.2, and no adjustment between the two chapters were done. Moreover, note that introducing codings varies across a wide range of wording (e.g., “0=no risk”, “No risk=0”, “code 0” and so on). Please keep a consistent writing style throughout the whole manuscript.
  13. Chapter 3 Results (page 6, line 298): As mentioned above, the presentation of the ANOVA was nearly perfect (except the bar charts, chapter 3.1, page 6, line 298), the presentation of chi-squared tests and logistic regression analyses (chapter 3.3., page 9, line 385) was very confusing. As a statistician, I appreciate that authors report all necessary statistical values in chapter 3.1 (page 6, line 298): F value, dfs, p-values, and a measure of effect size, this was great! However, in regards to consistency, the presentation of logistic regressions was the complete opposite: no p-values, no 95 % confidence intervals for ORs and so on. Honestly, I was very confused, because often the range of ORs was so large that I could not identify protective or risk factors for depression and anxiety (e.g., “OR ranged from 0.39 in Ukraine to 5.58 in Slovenia”, page 9, line 399). Perhaps, it would be useful to insert a new Table with nine columns (representing the nine countries) and stratify the presentation of results on country-level. Each line could then represent one outcome variable. Symbols could be used to identify protective factors or risk factors. For readers’ convenience, a Heat-Map could be useful to get a colored-driven overview. However, in the current version of the paper, the authors report empirical results per each outcome variable across all countries successively what was very confusing (e.g., “OR ranged from 0.39 in Ukraine to 5.58 in Slovenia”, page 9, line 399). By doing so, it remains completely open, whether this specific predictor variable is a protective factor or a risk factor. What is the distribution across countries (e.g., 4 times risk factor vs. 5 times protective factor)? Hence, I would recommend to use Heat Maps.
  14. Chapter 3.3 logistic regression (page 9, line 385): Why did the authors compute and report both, ORs and AORs? Note, later in the discussion, they rely on AORs only (“Therefore, we will focus on the multivariate models of mental health prediction in the discussion part.”, page 12, line 565). I would recommend that authors report and use AORs only because ORs are not adjusted for intercorrelations between predictor variables and, thus, are useless.
  15. Chapter 3.3 logistic regression (page 9, line 385): Why did the authors not include interaction terms in their multivariate logistic regression models? The sample size (N=2,349) is large so that more sophisticated models can be calculated.
  16. Chapter 3.3 logistic regression (page 9, line 385): Which method did the authors use for their multivariate logistic regression models for calculating AORs? No selection method? Backward selection? Forward selection? Stepwise? The authors should explain their statistical research methods in more detail (i.e., in chapter 2.4 Statistical Analysis, page 6, line 272).
  17. Chapter 3.3 logistic regression (page 9, line 385): The authors missed to report 95 % Confidence Intervals in their supplement Tables S3 and S4 for ORs and AORs.
  18. Chapter 4.4 Limitations (page 13, line 627): I appreciate that the authors discuss several limitations of their cross-sectional online survey study. However, they missed that they collected and analyzed a self-selected sample, meaning that participants could choose on their own whether to take part or not (i.e., no randomized sample possible). Hence, the analyzed sample could consist of “special” University students who admit to suffer from depression and anxiety. It is known from the “Transtheoretical Model of Health Behavior” (Prochaska & DiClemente, 2005) that patients could be classified by “Stages of Change” (SoC). The model claims 6 SoC (“Precontemplation”, “Contemplation”, “Preparation”, “Action”, “Maintenance”, and “Termination”). Hence, it is possible that self-selection biased the study sample and that measured depression and anxiety values in each country do not represent the truth. Only University students who are (1) aware of their depression/anxiety and (2) admit their depression/anxiety took part in this study. This would equal students placed in the SoC named “Preparation” (i.e., awareness of mental disease and planning to change it). Hence, I would urge the authors to add a self-selected study sample to their list of study limitations.

Author Response

Response to Editor and Reviewer 2 

We want to thank the Reviewer for the highly detailed evaluation of our manuscript and constructive comments. We do appreciate the valuable contribution and believe that it helped to improve our manuscript. We have addressed and incorporated all of the Reviewers’ comments. Our response is marked in italic.

Please see the provided changes in the attached manuscript. 

Response to Reviewer 2

  1. The reviewed paper deals with mental health of young adults during the COVID-19 pandemic. In particular, the authors of this paper present a cross-sectional online survey study in which depression and anxiety among University students in nine countries are compared. Cross-national data collection took place during the first wave of the coronavirus disease in the spring of 2020. After presentation of descriptive statistics, the authors used inferential methods (e.g., ANOVA) to find differences between the nine countries. After that, associations between depression/anxiety risks and other variables were analyzed with the two-dimensional chi-squared test for independency. The authors identified a large number of risk factors for depression and anxiety among University students with logistic regression analyses. Interestingly, these results were not homogenous but rather varied across countries. Thus, the authors concluded that planning and implementation of psychological intervention programs for students should include a country differentiation concerning mental health risk factors. I really enjoyed reading this article and thank the authors for the opportunity to review their interesting paper. Overall, I believe this is a very important paper in the current health crisis. Due to the COVID-19 pandemic, a lot of psychological health issues (e.g., depression, loneliness, anxiety) arose in the human population in the last year. I think it is worth to shed more light on this dramatic situation, not only from the physical health perspective (e.g., fever, dry cough, tiredness) but also from a psychological perspective. In the face of the worldwide COVID-19 pandemic it makes absolutely sense to study and compare depression and anxiety across different countries. I appreciate authors’ efforts in conducting a large cross-national online survey study in nine countries.
    However, I identified several issues in the current version of this paper. Some are theoretical in nature others are related to presented statistical analyses. Some are minor others are major. Below I listed some comments/suggestions that I hope will help the authors to further develop this line of work. So, let’s go through this:
  • Thank you for your comment and in-depth understanding of our main aim in this cross-national study.
  1. General comment: One of my major concerns about this paper are several serious breaks and flaws in writing style. Some parts of the paper were really good and nice to read, other parts were absolutely confusing (e.g., presentation of logistic regression analyses, chapter 3.3, page 9, line 385). I guess, the authors split up the paper and some persons are responsible for writing special parts of the paper (e.g., theoretical background, presentation of empirical results and so on). However, as a reviewer and reader, I felt this missing homogeneity so much.
    2a. For example, the order of measurements in the abstract (page 1, line 39) did not match the order of measurements in the measurement section (chapter 2.2, page 3, line 145).
    • Thank you for your comment. We have introduced changes to the manuscript: (1, lines 39-42)“Patient Health Questionnaire (PHQ-8), Generalized Anxiety Disorder (GAD-7), Exposure to COVID-19 (EC-19), Perceived Impact of Coronavirus (PIC) on students’ well-being, Physical Activity (PA), General Self-Reported Health (GSRH).”
    • and changed the presentation of the measurements so it would be consequent throughout the manuscript (5, lines 210-232).

2b.The presentation of the ANOVA was nearly perfect (chapter 3.1, page 6, line 298),

  • Thank you for this remark.

 2c. the presentation of chi-squared tests and logistic regression analyses (chapter 3.3., page 9, line 385) was very confusing.

  • According to the suggestion, we have rewritten and thoroughly presented the results of a Pearson’s χ2 test of independence (p.9-10, lines 374-457):

”The relationship between depression and gender was significant in Colombia (χ2(1) = 4.44, p < 0.05, ϕ = 0.17), Poland (χ2(1) = 7.28, p < 0.01, ϕ = 0.16), Russia (χ2(1) = 10.24, p < 0.01, ϕ = 0.19), Turkey (χ2(1) = 15.40, p < 0.001, ϕ = 0.22), and Ukraine (χ2(1) = 9.02, p < 0.01, ϕ = 0.17). Place of residence was not significantly associated with depression at all in any country. The level of study was significantly related to depression in Colombia (χ2(1) = 4.16, p < 0.05, ϕ = -0.16), Czechia (χ2(1) = 5.71, p < 0.05, ϕ = -0.13), and Slovenia (χ2(1) = 8.24, p < 0.01, ϕ = -0.19), while it was of no importance in the other countries. The relationship between depression and exposure to COVID-19 was significant in most countries (except Turkey and Colombia), including Slovenia (χ2(1) = 0.23, p < 0.001, ϕ = 0.33), Czechia (χ2(1) = 11.44, p < 0.001, ϕ = 0.19), Germany (χ2(1) = 4.16, p < 0.05, ϕ = 0.12), Poland, (χ2(1) = 7.60, p < 0.01, ϕ = 0.16), Ukraine (χ2(1) = 6.65, p < 0.01, ϕ = 0.15), Russia (χ2(1) = 10.39, p < 0.01, ϕ = 0.19), and Israel (χ2(1) = 15.99, p < 0.001, ϕ = 0.28). Except Colombia, the total PIC was linked to depression in the following: Slovenia (χ2(1) = 33.96, p < 0.001, ϕ = 0.40), Czechia (χ2(1) = 16.10, p < 0.001, ϕ = 0.23), Germany (χ2(1) = 39.80, p < 0.001, ϕ = 0.39), Poland, (χ2(1) = 20.85, p < 0.001, ϕ = 0.26), Ukraine (χ2(1) = 13.00, p < 0.001, ϕ = 0.21), Russia (χ2(1) = 19.72, p < 0.001, ϕ = 0.26), Turkey (χ2(1) = 9.26, p < 0.01, ϕ = 0.17), and Israel (χ2(1) = 35.64, p < 0.001, ϕ = 0.42). Qualifications were insignificant in Czechia and Poland. However, they were of concern in Slovenia (χ2(1) = 32.67, p < 0.001, ϕ = 0.40), Germany (χ2(1) = 16.95, p < 0.001, ϕ = 0.25), Ukraine (χ2(1) = 4.96, p < 0.05, ϕ = 0.13), Russia (χ2(1) = 15.66, p < 0.001, ϕ = 0.23), Turkey χ2(1) = 6.09, p < 0.05, ϕ = 0.14), Israel (χ2(1) = 12.94, p < 0.001, ϕ = 0.26), and Colombia (χ2(1) = 3.93, p < 0.05, ϕ = 0.16). Deterioration of economic status was a source of concern in most countries (except Poland): Slovenia (χ2(1) = 10.10, p < 0.01, ϕ = 0.22), Czechia (χ2(1) = 8.46, p < 0.01, ϕ = 0.16), Germany (χ2(1) = 14.02, p < 0.001, ϕ = 0.23), Ukraine (χ2(1) = 4.65, p < 0.05, ϕ = 0.12), Russia (χ2(1) = 11.25, p < 0.001, ϕ = 0.20), Turkey χ2(1) = 13.58, p < 0.001, ϕ = 0.21), Israel (χ2(1) = 13.12, p < 0.001, ϕ = 0.26), and Colombia (χ2(1) = 5.06, p < 0.05, ϕ = 0.18). Relationships with friends and family members were a source of concern in most countries during the COVID-10 pandemic (except Colombia): Slovenia (χ2(1) = 13.70, p < 0.001, ϕ = 0.27), Czechia (χ2(1) = 11.06, p < 0.001, ϕ = 0.19), Germany (χ2(1) = 9.76, p < 0.01, ϕ = 0.19), Poland (χ2(1) = 24.57, p < 0.05, ϕ = 0.29), Ukraine (χ2(1) = 8.36, p < 0.01, ϕ = 0.16), Russia (χ2(1) = 24.16, p < 0.001, ϕ = 0.29), Turkey χ2(1) = 4.26, p < 0.05, ϕ = 0.12), and Israel (χ2(1) = 14.38, p < 0.001, ϕ = 0.27). The relationship between high depression and insufficient physical activity (PA less than 150 min. per week) was revealed in Poland (χ2(1) = -4.85, p < 0.05, ϕ = -0.13), Ukraine (χ2(1) = -11.77, p < 0.001, ϕ = -0.20), Russia (χ2(1) = -7.36, p < 0.01, ϕ = -0.16), and Israel (χ2(1) = 3.89, p < 0.05, ϕ = -0.14). An association between high depression and worse physical health was significant in most countries (except Czechia and Ukraine): Slovenia (χ2(1) = 15.16, p < 0.001, ϕ = 0.27), Germany (χ2(1) = 21.71, p < 0.001, ϕ = 0.29), Poland (χ2(1) = 13.14, p < 0.001, ϕ = 0.21), Russia (χ2(1) = 10.00, p < 0.01, ϕ = 0.19), Turkey χ2(1) = 5.89, p < 0.05, ϕ = 0.14), Israel (χ2(1) = 4.89, p < 0.05, ϕ = 0.16), and Colombia (χ2(1) = 5.14, p < 0.05, ϕ = 0.18). When students compared self-rated physical health to other people of the same age, the association between depression and comparative health was noted in most countries (except Russia): Slovenia (χ2(1) = 20.88, p < 0.001, ϕ = 0.32), Czechia (χ2(1) = 14.45, p < 0.001, ϕ = 0.22), Germany (χ2(1) = 7.09, p < 0.01, ϕ = 0.16), Poland (χ2(1) = 21.64, p < 0.001, ϕ = 0.27), Ukraine (χ2(1) = 6.96, p < 0.01, ϕ = 0.15), Turkey χ2(1) = 10.73, p < 0.001, ϕ = 0.19), Israel (χ2(1) = 5.76, p < 0.05, ϕ = 0.17), and Colombia (χ2(1) = 6.83, p < 0.01, ϕ = 0.21).

Anxiety was related to gender in Israel (χ2(1) = 4.87, p < 0.05, ϕ = 0.16), Russia (χ2(1) = 4.15, p < 0.05, ϕ = 0.12), Turkey (χ2(1) = 9.15, p < 0.01, ϕ = 0.17) and Ukraine (χ2(1) = 7.52, p < 0.01, ϕ = 0.16). An association between anxiety and place of residence was significant solely in Poland (χ2(1) = 7.67, p < 0.01, ϕ = 0.16). The relationship between the level of study and anxiety was observed in Czechia (χ2(1) = 6.80 p < 0.01, ϕ =-0.15) and Slovenia (χ2(1) = 5.61, p < 0.05, ϕ = -0.16). Exposure to COVID-19 was significantly associated with anxiety in all countries, Slovenia (χ2(1) = 13.25, p < 0.001, ϕ = 0.25), Czechia (χ2(1) = 10.34, p < 0.01, ϕ = 0.18), Germany (χ2(1) = 8.82, p < 0.01, ϕ = 0.18), Poland (χ2(1) = 8.97, p < 0.01, ϕ = 0.17), Ukraine (χ2(1) = 10.03, p < 0.01, ϕ = 0.18), Russia (χ2(1) = 6.95, p < 0.01, ϕ = 0.16), Turkey χ2(1) = 7.90, p < 0.01, ϕ = 0.16), Israel (χ2(1) = 10.28, p < 0.01, ϕ = 0.23), and Colombia (χ2(1) = 4.40, p < 0.05, ϕ = 0.17). The total PIC was significantly related to anxiety in all countries: Slovenia (χ2(1) = 29.98, p < 0.001, ϕ = 0.38), Czechia (χ2(1) = 13.01, p < 0.001, ϕ = 0.20), Germany (χ2(1) = 9.36, p < 0.01, ϕ = 0.18), Poland (χ2(1) = 12.74, p < 0.001, ϕ = 0.21), Ukraine (χ2(1) = 17.48, p < 0.001, ϕ = 0.24), Russia (χ2(1) = 5.34, p < 0.05, ϕ = 0.14), Turkey χ2(1) = 11.92, p < 0.001, ϕ = 0.20), Israel (χ2(1) = 32.27, p < 0.001, ϕ = 0.40), and Colombia (χ2(1) = 8.14, p < 0.01, ϕ = 0.23). Qualifications (graduation) were an important source of concern in most countries (except in Poland): Slovenia (χ2(1) = 23.19, p < 0.001, ϕ = 0.33), Czechia (χ2(1) = 17.80, p < 0.001, ϕ = 0.24), Germany (χ2(1) = 11.62, p < 0.001, ϕ = 0.21), Ukraine (χ2(1) = 11.31, p < 0.001, ϕ = 0.19), Russia (χ2(1) = 7.37, p < 0.01), Turkey χ2(1) = 8.78, p < 0.01, ϕ = 0.27), Israel (χ2(1) = 9.81, p < 0.01, ϕ = 0.22), and Colombia (χ2(1) = 8.06, p < 0.01, ϕ = 0.23). Deterioration of economic status was important in Slovenia (χ2(1) = 4.96, p < 0.05, ϕ = 0.15), Czechia (χ2(1) = 15.81, p < 0.001, ϕ = 0.23), Germany (χ2(1) = 4.27, p < 0.05, ϕ = 0.13), Russia (χ2(1) = 4.47, p < 0.05, ϕ = 0.13), Turkey χ2(1) = 15.00, p < 0.001, ϕ = 0.22), and Israel (χ2(1) = 4.42, p < 0.05, ϕ = 0.15). Excluding Germany, social relationships were of importance in Slovenia (χ2(1) = 13.39, p < 0.001, ϕ = 0.25), Czechia (χ2(1) = 4.27, p < 0.05, ϕ = 0.12), Poland (χ2(1) = 21.95, p < 0.001, ϕ = 0.27), Ukraine (χ2(1) = 13.05, p < 0.001, ϕ = 0.21), Russia (χ2(1) = 10.04, p < 0.01, ϕ = 0.19), Turkey χ2(1) = 4.87, p < 0.05, ϕ = 0.13), Israel (χ2(1) = 14.66, p < 0.001, ϕ = 0.27), and Colombia (χ2(1) = 5.62, p < 0.01, ϕ = 0.19). The relationship between the insufficient level of physical activity and high anxiety was statistically significant in Poland (χ2(1) = -7.94, p < 0.01, ϕ = 0.16) and Ukraine (χ2(1) =- 4.80, p < 0.05, ϕ = 0.12). Poor physical health was related to high anxiety in most countries (except Czechia and Israel): Slovenia (χ2(1) = 6.45, p < 0.05, ϕ = 0.18), Germany (χ2(1) = 9.38, p < 0.01, ϕ = 0.17), Poland (χ2(1) = 18.51, p < 0.001, ϕ = 0.25), Ukraine (χ2(1) = 6.47, p < 0.05, ϕ = 0.14), Russia (χ2(1) = 4.97, p < 0.05, ϕ = 0.13), Turkey χ2(1) = 13.29, p < 0.001, ϕ = 0.21), and Colombia (χ2(1) = 4.24, p < 0.05, ϕ = 0.16). Comparative physical health was linked to anxiety in most countries (except Ukraine, Israel): Slovenia (χ2(1) = 10.75, p < 0.01, ϕ = 0.23), Czechia (χ2(1) = 4.64, p < 0.05, ϕ = 0.12), Germany (χ2(1) = 4.19, p < 0.05, ϕ = 0.13), Poland (χ2(1) = 18.56, p < 0.001, ϕ = 0.25), Russia (χ2(1) = 4.50, p < 0.05, ϕ = 0.13), Turkey χ2(1) = 11.19, p < 0.001, ϕ = 0.19), and Colombia (χ2(1) = 8.00, p < 0.01, ϕ = 0.23).”

2d. Again, I think this paper has big potential, especially the cross national data collection, but large parts of this article must be rewritten and re-structured before a publication of this article makes sense.

  • We can see your point. The manuscript has been significantly changed. We have restructured the methodology part. We have excluded the bivariate regression models and rewritten all of the results with a new visual presentation of the results (box plots and heat Maps). The bias-corrected accelerated bootstrapping (BCa) method of estimation regression coefficient was also applied, with the number of replications is set to 5,000 for each country. The goodness of fit of the regression model was assessed using pseudo R2, including Cox and Snell R2CS, McFadden R2McF, and Nagelkerke R2N.
  • We have also partially modified the introduction and discussion.

3. Theoretical background: A space is missing behind the dot and the word “Despite” („period [18].Despite“, page 2, line 77). Please add a space.

  • Thank you for this detailed remark. We have added a space.

4. Theoretical background: “Mental health issues are common in the student population— more than one-third of students experienced some form of mental health problems in the pre-pandemic period [18].” (page 2, line 75 to 77). I did not get the point, why University students are so prone for depression and anxiety. The same is true for other populations (e.g., pupils). Why do the authors focus on this special population of University students? They cite a lot of literature in this context but missed to clarify their arguments. It is not sufficient that readers must search for themselves to get an appropriate answer to that question. I can formulate some hypotheses on my own (e.g., living on their own, no or reduced contact to colleagues, no learning groups, no financial income) but also for the opposite (e.g., living by parents, online learning platforms). Hence, I urge the authors to add some passages to the theoretical background that explain and clarify the vulnerability of University students for depression and anxiety.

  • Thank you for noticing this. We agree with the undertaken issue and we provided changes in the introduction:
    • (p.2, 84-90) “Financial difficulties constitute the risk factor for the increase of anxiety and depression levels. They can also lead to poor academic performance [23]. Financial concerns are not the only factor affecting students’ mental health issues. They can also be influenced by [24] academic pressure and demanding workloads [25], student mistreatment and abuse [26] and worry about own health [27]. Students are particularly susceptible to affective disorders due to high social expectations as they are deemed to represent the future of a community [28].”;
    • (p.2, lines 92-97) “Social isolation during the COVID-19 pandemic revealed a higher experience of insecurity concerning housing and employment opportunities [33], smaller living space and lower levels of social interactions in young adults compared to adults [4,34]. Academic stress and virtual learning are the crucial risk factor as well [35,36]. According to the International Labor Organization [37], the education sector has been strongly affected by the COVID-19 pandemic.”.

5. Theoretical background (page 3, line 118): not “CIOVID” but rather “COVID”

    • Thank you, we have provided the change.

6. Chapter 2.2 Measurements: As mentioned in my point 1, the order of measurements in the abstract (PHQ-8, GAD-7, […], page 1, line 39) do not match the order of measurements in the measurement section (GAD-7, PHQ-8, […], chapter 2.2, page 3, line 145). Please keep a consisted writing style and correct this, so that the order of measurements is constant throughout the whole manuscript.

  • Yes, thank you for noticing this. We have changed all of the measurements presentations throughout the abstract, (1, lines 39-42)“Patient Health Questionnaire (PHQ-8), Generalized Anxiety Disorder (GAD-7), Exposure to COVID-19 (EC-19), Perceived Impact of Coronavirus (PIC) on students’ well-being, Physical Activity (PA), General Self-Reported Health (GSRH)”.
  • Measurement part (4-8, lines 208-230) and presentation in the figures and tables.

7. Chapter 2.2 Measurements (page 4, line 169): It seems that the instrument “Exposure to COVID-19” is self-constructed by the authors. Please add this hint or, in the other case, cite appropriate literature.

  • We have changed the place of the reference to make it more readable: (p. 5, line 230) “Exposure to COVID-19 [39]”.

8. Chapter 2.2 Measurements (page 4, line 182): It seems that the instrument “Perceived Impact of Coronavirus (PIC)” is self-constructed by the authors. Please add this hint or, in the other case, cite appropriate literature.

  • We have changed the place of the reference as to make more readable: (p. 5, line 243) “The Perceived Impact of Coronavirus (PIC) on students’ well-being [39]”.

9. Chapter 2.2 Measurements (page 5, line 222): Here is a good example of missing consistency throughout the whole manuscript. The authors add behind the Cronbach’s α value the (reduced) sample size (N=2,349). However, at this point the reader can’t know that the authors recruited 2,453 respondents but 104 persons declined participation, resulting in a final sample size of 2,349 (see chapter 2.3 Participants, page 5, line 233). This information is presented in a later chapter so that I claim an infrigment of consistency which probably occurred of fractured writing. Probably, Person A wrote chapter 2.2, Person B wrote chapter 2.3 and no adjustment between the two chapters occurred. Thus, I urge the authors to correct this and keep a consistent writing style.

    • We agree with this remark. We have started the Materials and Methods section from the participants’ subsection and the study design (p.3-4, lines 141-186). This allowed for the presentation of the recruited total research sample in the first place.

10. Chapter 2.3 Participant (page 6, line 266): “The sample size for each country was computed by the means of G*Power software”. As a statistician, I appreciate very much that the authors did a sample size calculation a-priori before the recruitment started. However, they missed to present the actual values (significance level α, assumed power, and effect size) to calculate the minimum number of participants per each country needed. In scientific papers, it is important to replicate the calculated sample size for transparency reasons. How did the authors get the number 159? The authors report the underlying statistical test (chi-squared test), hint to the significance level α (p<0.05 and 95% confidence intervals -> α seems to be 5%), and the two-tailed nature of the test only. However, which statistical power and which effect size did they use for the sample size calculation and why? Is there any rationale for the chosen (but not presented) effect size, any literature that could be used for strengthen the assumed effect size? I urge the authors to rewrite this section and report all necessary values.

    • Thank you for this remark. We have changed this part to show in detail the sample size calculation: (p.3, lines 142-146) “The required sample size for each country group was computed a priori by means of G*Power software [61]. In order to obtain a medium effect size of Cohen’s W = 0.03 with given 95% power in a 2 x 2 χ2 contingency table, df = 1 (two groups in two categories each, two tailed), α = 0.05, G*Power suggests 145 participants are required in each country group (non-centrality parameter λ = 13.05, critical χ2 = 3.84, power = 0.95).”

11. Chapter 3.1 (page 7, line 329): Here, the authors illustrate measured depression and anxiety across the nine countries with bar charts (Figure 1 and Figure 2). Although this is often done in scientific papers, it is still wrong. Why? Bar charts represent mean values which is one value only, a dot, and not a bar. Hence, bar charts make no sense for illustrating mean values. As a statistician, I would recommend using box plots (in full word: box-whisker plots) instead. Box plots are in this situation more appropriate than bar charts, because they represent not one single value but rather the distribution of measured values. Using box plots has the advantage that the reader gets an idea about the range of values in the sense of transparent distribution. Median, first and third quantile as well as minimum and maximum values, even outliers can be detected within box plots. Hence, I urge the authors to replace the bar charts and use box plots instead for illustrating their measured depression and anxiety values.

    • Thank you. We appreciate this comment. We have provided new Figures using the box plots (p.8, lines 359-364).

12. Chapter 3.1 (page 7, line 329): Something strange happened to Footnotes of Figure 1 and Figure 2 (smaller line spacing in Figure 2 compared to Figure 1). Please adjust this.

    • Yes, we provided changes (p.8, lines 359-364).

13. Chapter 3.2 (page 8, line 342): Here is another good example of missing consistency throughout the whole manuscript. An unnecessary repetition occurs in this section. As a statistician, I really appreciate that authors clearly describe their codings (e.g., “coded No Risk = 0”, page 8, line 342). However, they already introduced their codings in chapter 2.2 Measurements (page 3, line 146). I guess, it is the same problem as before: Person A wrote chapter 2.2, Person B wrote chapter 3.2, and no adjustment between the two chapters were done. Moreover, note that introducing codings varies across a wide range of wording (e.g., “0=no risk”, “No risk=0”, “code 0” and so on). Please keep a consistent writing style throughout the whole manuscript.

    • Thank you for noticing this. We havechanged all of the codes’ descriptions:
  • (p.5, line 216) “0 = No risk (PHQ-8 < 10), 1 = Risk (PHQ-8 > 10)”;
  • (p.5, lines 228-229) “0 = No risk (GAD-7 < 10), 1 = Risk (GAD-7 > 10)”; (p.5, line 241) “(0 = No, 1 = Yes)”;
  • (p.5, lines 240-242) “The results were divided into two categories for the χ2 independence test and logistic regression: 0 = Low exposure (score 0), 1 = High exposure (scores 1-8)”;
  • (p.5-6, lines 254-259) “The total PIC was coded as follows (for the χ2 independence test and logistic regression): 0 = Lower (PIC < 15), 1 = Higher (PIC > 16). We added scores of items PIC1 and PIC2 for the Qualifications scale and then coded as 0 = Low (scores 2-6), 1 = High (scores 7-10). Social Relationships scale consisted of items PIC4 and PIC5 coded as 0 = Low (scores 2-4), 1 = High (scores 5-10). Single item PIC3 concerning Economic Status scale was coded as 0 = Low (scores 1-3), 1 = High (scores 4-5).”;
  • (p.6, lines 270-271) “0 = Sufficient (PA > 150 minutes weekly) and 1 = Insufficient (PA < 150 minutes weekly)”;
  • (p.6, lines 281-283) “0 = Better health (GSRH < 3), 1 = Worse health (GSRH > 4)”;
  • (p.6, 288) “0 = village and town, 1 = city, agglomeration, or metropolis”
  • (p.6, lines 291) “0 = Bachelor and 1 = Master (for Master or higher)”.

14. Chapter 3 Results (page 6, line 298): As mentioned above, the presentation of the ANOVA was nearly perfect (except the bar charts, chapter 3.1, page 6, line 298), the presentation of chi-squared tests and logistic regression analyses (chapter 3.3., page 9, line 385) was very confusing. As a statistician, I appreciate that authors report all necessary statistical values in chapter 3.1 (page 6, line 298): F value, dfs, p-values, and a measure of effect size, this was great! However, in regards to consistency, the presentation of logistic regressions was the complete opposite: no p-values, no 95 % confidence intervals for ORs and so on. Honestly, I was very confused, because often the range of ORs was so large that I could not identify protective or risk factors for depression and anxiety (e.g., “OR ranged from 0.39 in Ukraine to 5.58 in Slovenia”, page 9, line 399). Perhaps, it would be useful to insert a new Table with nine columns (representing the nine countries) and stratify the presentation of results on country-level. Each line could then represent one outcome variable. Symbols could be used to identify protective factors or risk factors. For readers’ convenience, a Heat-Map could be useful to get a colored-driven overview. However, in the current version of the paper, the authors report empirical results per each outcome variable across all countries successively what was very confusing (e.g., “OR ranged from 0.39 in Ukraine to 5.58 in Slovenia”, page 9, line 399). By doing so, it remains completely open, whether this specific predictor variable is a protective factor or a risk factor. What is the distribution across countries (e.g., 4 times risk factor vs. 5 times protective factor)? Hence, I would recommend to use Heat Maps.

    • We have used Heat Maps for presenting multivariate logistic regression estimates: (p. 12, lines 541-546) Figure 3, (p.14, lines 609-614) Figure 4. In accordance with a subsequent comment, we have excluded the bivariate logistic regression models from the manuscript.

 15. Chapter 3.3 logistic regression (page 9, line 385): Why did the authors compute and report both, ORs and AORs? Note, later in the discussion, they rely on AORs only (“Therefore, we will focus on the multivariate models of mental health prediction in the discussion part.”, page 12, line 565). I would recommend that authors report and use AORs only because ORs are not adjusted for intercorrelations between predictor variables and, thus, are useless.

    • Thank you for this suggestion. As noted earlier in response to comment 2d, we have provided changes according to this comment and excluded the bivariate regression models. We have also rewritten all of the results with a new visual presentation of the results (Heat Maps) (p.10-14, lines 458-614). We have also added the bias-corrected accelerated bootstrapping (BCa) method of estimation regression coefficient was also applied, with the number of replications is set to 5,000  for each country. The goodness of fit of the regression model was assessed using pseudo R2, including Cox and Snell R2CS, McFadden R2McF, and Nagelkerke R2N. We have also provided changes in the discussion section in accordance to the multivariate regression models and BCa results:
    • (p.16, lines 713-719) ”The most frequent predictors of depression and anxiety in the nine countries were gender, exposure to COVID-19, and comparative physical health.The multiple logistic regression proved gender to be a significant predictor of anxiety but only among Israeli, Ukrainian, and Turkish students. Gender was a more frequent predictor of depression among Colombian, Polish, Russian, Turkish, and Ukrainian students, but a less significant predictor in Colombia.”;
  • (p.16-17, lines 742-743) “for depression, and in four countries (Czechia, Poland, Turkey, and Ukraine)”;
  • (p.17, lines 753-754) “The perceived impact of COVID-19 on students’ economic status was significantly associated with depression and anxiety in the majority of the countries, as the above analysis showed.”;
  • (p.17, lines 756-761) “However, when the economic status was introduced in multiple models, it turned out to be a trivial predictor of depression while being a significant predictor of anxiety only in one country (Czechia). Therefore, even though PIC Economic Status is relevant when analyzed as a singular risk factor for mental health, when combined with other risk factors for mental health, such as exposure to COVID-19 or female gender, it becomes insignificant”;
  • (p.17, lines 766-767) “in Russia and Ukraine”;
  • (p.17, lines 771-774) “(Czechia, Poland, Slovenia, and a weaker predictor in Turkey). For anxiety, this was true merely for two countries. However, the results in Colombia and Czechia were not confirmed when the bias-corrected accelerated bootstraping method was introduced”.

 16. Chapter 3.3 logistic regression (page 9, line 385): Why did the authors not include interaction terms in their multivariate logistic regression models? The sample size (N=2,349) is large so that more sophisticated models can be calculated.

    • Thank you for this suggestion. However, the aim of this paper was to show the results in each country separately. Considering the number of analyses for depression and anxiety in each country, we believe that introducing another statistical analysis would make it even more unclear. 

17. Chapter 3.3 logistic regression (page 9, line 385): Which method did the authors use for their multivariate logistic regression models for calculating AORs? No selection method? Backward selection? Forward selection? Stepwise? The authors should explain their statistical research methods in more detail (i.e., in chapter 2.4 Statistical Analysis, page 6, line 272).

    • Thank you for this suggestion. We have added an explanation: (p.7, lines 316-317)  “All predictors were entered into the model simultaneously”.

18. Chapter 3.3 logistic regression (page 9, line 385): The authors missed to report 95 % Confidence Intervals in their supplement Tables S3 and S4 for ORs and AORs.

  • Thank you for pointing this out. We have introduced all needed statistics to describe the multivariate logistic regression analysis for each of the nine countries: (p.7, lines 316-325) “All predictors were entered into the model simultaneously. The following statistics were calculated for estimation: coefficient estimates, 95% confidence intervals (CI) for the regression coefficient, standard errors of the regression coefficient, odds ratio, z-values, and their corresponding p-values. The bias-corrected accelerated bootstrapping (BCa) method of estimating regression coefficient was also applied, with the number of replications set to 5,000 (if the bias-corrected 95% confidence intervals (CIB) did not include the null value, then a statistically significant effect was considered). Goodness of fit of the regression model was assessed using pseudo R2, including Cox and Snell R2CS, McFadden R2McF, and Nagelkerke R2N. All analyses were performed using Statistica 13.3 [68] and JASP [69]”.
  • The updated Tables are included in the Supplement and in the manuscript (p.10-14).

19. Chapter 4.4 Limitations (page 13, line 627): I appreciate that the authors discuss several limitations of their cross-sectional online survey study. However, they missed that they collected and analyzed a self-selected sample, meaning that participants could choose on their own whether to take part or not (i.e., no randomized sample possible). Hence, the analyzed sample could consist of “special” University students who admit to suffer from depression and anxiety. It is known from the “Transtheoretical Model of Health Behavior” (Prochaska & DiClemente, 2005) that patients could be classified by “Stages of Change” (SoC). The model claims 6 SoC (“Precontemplation”, “Contemplation”, “Preparation”, “Action”, “Maintenance”, and “Termination”). Hence, it is possible that self-selection biased the study sample and that measured depression and anxiety values in each country do not represent the truth. Only University students who are (1) aware of their depression/anxiety and (2) admit their depression/anxiety took part in this study. This would equal students placed in the SoC named “Preparation” (i.e., awareness of mental disease and planning to change it). Hence, I would urge the authors to add a self-selected study sample to their list of study limitations.

    • We have added this limitation to the manuscript: (p. 17, lines 786-787) “Another limitation is a self-selected study sample”.
    • However, we want to note that the invitation and the instruction regarding the survey were written in a positive manner, informing about the well-being during the COVID-19 pandemic and not directly about depression and anxiety. The participants were informed that they could stop fulfilling the questionnaires at any moment, and the participation is voluntary. Nonetheless, the response rate was very high (96%).

Round 2

Reviewer 2 Report

I thank the authors for submitting a revised version of their manuscript entitled “A Comparison of Depression and Anxiety among University Students in Nine Countries during the COVID-19 Pandemic”.

I applause, it was a real pleasure for me to read the revised version of this paper and I definitely pay great respect to the authors’ efforts. Honestly, the authors did a great job and re-wrote and re-arranged large parts of their manuscript. As a big movie fan, I would say that the authors provided a “director’s cut” of their original manuscript which is now more sophisticated and elaborated. In my opinion, the revised manuscript increased a lot in comparison to the first version. Now everything read more smoothly (i.e., consistency throughout the manuscript is given) and I could follow authors’ argumentations easily (e.g., why University students are prone to depression and anxiety). The statistical analyses were adjusted (e.g., no bivariate logistic regression models anymore; now bootstrapping method and 95% CIs included) and figures were changed (boxplots instead of bar charts, heat maps).

In summary, the authors have responded to all my concerns adequately. I have no further comments. Thus, I recommend this excellent manuscript for publication (accept in present form). Well done!